# Is Knowledge in Multilingual Language Models Cross-Lingually Consistent?

## Abstract

Few works study the variation and cross-lingual consistency of factual knowledge embedded in multilingual models. However, cross-lingual consistency should be considered to assess cross-lingual transferability, maintain the factuality of the model's knowledge across languages, and preserve the parity of language model performance. We are thus interested in analyzing, evaluating, and interpreting cross-lingual consistency for factual knowledge. We apply interpretability approaches to analyze a model's behavior in cross-lingual contexts, discovering that multilingual models show different levels of consistency, subject to either language families or linguistic factors. Further, we identify a cross-lingual consistency bottleneck manifested in middle layers. To mitigate this problem, we try vocabulary expansion, additional cross-lingual objectives, adding biases from monolingual inputs, multi-task fine-tuning, and code-switching training. We find that all these methods, except for multi-task fine-tuning, boost cross-lingual consistency to some extent, with cross-lingual supervision and code-switching training offering the best improvement.

## 1 Introduction

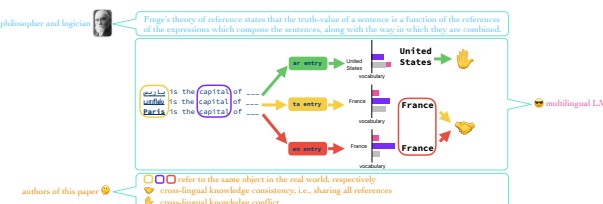

Figure 1: Illustration of Cross-lingual knowledge consistency. Frege's theory of reference defines the reference of a sub-sentential expression as the object singled out by the name.

Frege's theory of reference (Frege, 1892) indicates that the knowledge conveyed by a sentence depends on the references of the expressions that compose the sentence. A salient aspect of humanity is that while people may speak different languages, they can share common knowledge. Thus, references and knowledge should be consistent across languages, and a multilingual model serving as a knowledge base (Gupta & Srikumar, 2021; Kassner et al., 2021; Hu et al., 2024) should provide consistent knowledge when queried in different languages. Not only does this theory contribute to the cross-lingual performance and maintain the knowledge across languages, but it ensures parity and self-consistency of model performance (Hupkes et al., 2023; Wang et al., 2023). This motivates us to evaluate the knowledge consistency of multilingual language models across languages.

Although recent advances have shown that multilingual models are effective for cross-lingual transfer and generalization (Conneau et al., 2020; Xue et al., 2020; Hu et al., 2020; Muennighoff et al., 2023), Kassner et al. (2021); Fierro & Søgaard (2022); Qi et al. (2023) reported that the prediction varies from one language to others when recalling knowledge in different languages. For given parallel statements, a language model may output predictions for a particular query that differs from one obtained from the query's translation. This examination implicitly instantiates Frege's theory of reference to check knowledge consistency across languages that parallel statements with the same references for sub-sentential expressions such as entities should have the same knowledge.

Inspired by that theory, we hypothesize that multilingual language models recall consistent factual knowledge for coreferential statements in cross-lingual settings. To evaluate this hypothesis (Figure 1), we create code-mixed coreferential statements from monolingual statements by substituting a subject entity with an equivalent one in another language that shares the same reference and attempt to answer two questions: *1) do multilingual language models recall factual knowledge for the coreferential statements in a similar manner on different languages?* and *2) how does the mechanism of multilingual language models work on the incorporation between entities or references in cross-lingual settings?* Our study is related to a broader linguistic phenomenon of entity-level code-switching: an entity code-switches between two languages without changing the reference. More recently, we share a similar goal with knowledge incorporation and editing (Beniwal et al., 2024; Li et al., 2024), as we incorporate a coreferential entity from other languages for factual knowledge recall in cross-lingual settings. Our main findings are:

- Multilingual language models can leverage coreferential entities across similar languages, e.g., similar writing scripts, to maintain cross-lingual knowledge consistency when recalling factual knowledge but worse across dissimilar languages.

- Scaling models is not a promising strategy to improve cross-lingual consistency as we observed a bottleneck starting from middle layers across different model families and sizes.

- Consistency patterns in feed-forward neurons and subject–object attention scores are indistinguishable for similar languages and distinguishable for dissimilar languages.

Based on our main findings, we further evaluate several mitigations that could resolve inconsistency issues with the results described below, particularly those observed in dissimilar languages.

- There is a partial causality between adding monolingual biases and improving cross-lingual knowledge consistency. Thus, adding bias could be a potential method to calibrate consistency across languages while we do need to think a better way to incorporate such bias.

- Expanding the multilingual vocabulary, adding word alignment objective, and code-switching training can improve the cross-lingual consistency as such method helps in aligning coreferential entities across languages to alleviate the consistency bottleneck.

- Multi-task fine-tuning is not promising to benefit the cross-lingual consistency as it potentially refines specific attention heads for in-context information instead of knowledge.

Our contribution is to offer an understanding of multilingual language models' limitations under cross-lingual settings and highlight potential research directions to address such issues.

## 2 METHODOLOGY

### 2.1 TASK DEFINITION

We focus on a code-mixed context-independent cloze task. This setting forces the multilingual model to rely on its internal knowledge base and recall the common knowledge shared by coreferential entities across languages because of cross-lingual generalization [1]. In the following introduction, we will define the evaluation task mathematically. Let $I = \{S^{l1}, \cdots, O, \cdots\} \in l1$ [2] be a statement, where $l1$ stands for matrix language (the predominant language), $S^{l1} = \{s_1, \cdots, s_k\} \in l1$ are subject sub-tokens, and $O = \{o_1, o_2, \cdots, o_j\} \in l1$ denote object sub-tokens. This statement is used to create a masked input $I_{mono} = \{S^{l1}, \cdots, M, \cdots\}$, where $M = \{mask_1, \cdots, mask_j\}$ are the mask (or the sentinel token $M = <extra\_id\_0>$) used to substitute $O$ in $I$. We define n-gram prediction for the mask $O$, denoted as $Cand(O_{\in V}|I_{mono})$, as the top-k n-gram candidates with token lengths ranging from 1 to $j$ obtained from beam search decoding over the model's vocabulary $V$. Considering the trade-off between the computational time and the holistic of the language model's prediction, we set the top-k threshold and beam search width to 5. Similarly, we can create a code-mixed coreferential statement $I_{cm}$ by replacing $S^{l1}$ with a coreferential subject $S^{l2}$ in the embedded language $l2$ (the subsidiary language) in order to obtain $Cand(O_{\in V}|I_{cm})$. Therefore,

---

[1]See limitation in §6.

[2]The surface structure is not restricted. We use the common subject–object structure as an example.

$I_{cm}$ and $I_{mono}$ are coreferential and expected to recall the same knowledge. Finally, we define cross-lingual knowledge consistency as $0 \leq f_{metric}(Cand(O_{\in V}|I_{mono}), Cand(O_{\in V}|I_{cm})) \leq 1$, where $f_{metric}$ is a consistency metric defined in the next subsection. If $f_{metric} = 1$, it implies that multilingual language models recall factual knowledge for the coreferential statements $I_{mono}$ and $I_{cm}$ in an identical manner. The coreferential statements disagree if $f_{metric} = 0$. Note that we do not consider whether the prediction is correct. Instead, $f_{metric}$ evaluates the parity and consistency across the languages that the model is expected to output similar candidates for $I_{mono}$ and $I_{cm}$.

From a probability view, we can define our task as measuring the difference between two distributions, $Cand(O_{\in V}|I_{cm}) = P(O_{\in V}|K_{\theta})P(K_{\theta}|S^{l2}, I_{\backslash(S^{l1} \cap O)})$ and $Cand(O_{\in V}|I_{mono}) = P(O_{\in V}|K_{\theta}^{*})P(K_{\theta}^{*}|S^{l1}, I_{\backslash(S^{l1} \cap O)})$, where $K_{\theta}$ is the knowledge recalled from the model given the preceding context, and $I_{\backslash(S^{l1} \cap O)}$ stands for $I$ without both the subject and the object. Then, cross-lingual knowledge consistency between $K_{\theta}^{*}$ and $K_{\theta}$ reflects on the measured difference. The high-level idea of this evaluation task is illustrated in Figure 1 where en entry "Paris is the capital of ___" is evaluated with its possible code-mixed statements (ar entry & ta entry). In this example, $S^{l1}$, $I_{\backslash(S^{l1} \cap O)}$, and $S^{l2}$ are "Paris", "is the capital of", and the ar or ta entry for "Paris", respectively. If coreferential subject entries are trained to generalize across languages, we could observe the cross-lingual consistency. In addition, we are aware of a baseline from this probability view. Specifically, we define the baseline as the difference between $Cand(O_{\in V}|I_{mono})$ and $Cand(O_{\in V}|I_{\backslash(S^{l1} \cap O)}) = P(O_{\in V}|K_{\theta}^{\alpha})P(K_{\theta}^{\alpha}|I_{\backslash(S^{l1} \cap O)})$, measuring agnostic consistency without the coreferential subjects $S^{l1}$ and $S^{l2}$ in cross-lingual settings. In implementation, we mask the both subject and object entities to create the "code-mixed" counterpart as the baseline. Readers can refer to Appendix §A.1 for our implementation.

## 2.2 Metric Function and Interpretability Approach

Readers can refer to Appendix §A.2 for more details, e.g., equations.

**Consistency Metrics.** For $f_{metric}$, **Top@1 Accuracy** and **RankC** (i.e., weighted Precision@5) (Qi et al., 2023) are used to evaluate the cross-lingual knowledge consistency between $Cand(O_{\in V}|I_{mono})$ and $Cand(O_{\in V}|I_{cm})$. Since we observe similar experimental results on Top@1 and RankC, Top@1 results are moved to Appendix §A.3.

**Consistency Evolution.** We analyze the "evolution" of consistency scores as the layer goes deeper to trace the consistency bottleneck and understand the models' behavior. Since the encoder part is crucial for both encoder and encoder-decoder language models to understand the input, we apply LogitLens (nostalgebraist, 2019) (for the xlm-r family) and DecoderLens (Langedijk et al., 2023) (for the mT0 family) to each encoder layer to obtain the layer-wise distribution $O_{\in V}$.

**Subject–Object Attention.** Inspired by the attention weight analysis method (Clark et al., 2019), we calculate the sum of all subject tokens' attention scores across all masked tokens and average those scores over all possible $I_{mono}$ and $I_{cm}$ statements. Then we collect the difference between average attention scores of $I_{cm}$ and those obtained from $I_{mono}$. Note that masked tokens are used to prompt the corresponding object tokens of the masked statements, as defined in our task definition.

$IG^2$ **Score** We do minor modification on $IG^2$ (Liu et al., 2024) to measure the impact of each feed-forward neuron on the logits of the mask tokens where the higher the value is, the more critical the neuron is to predict the ground truth object on the mask tokens.

## 2.3 Dataset and Model

**Dataset.** We use mLAMA dataset (Kassner et al., 2021) that provides parallel triples (object, predicate, subject) in 53 languages written in cloze task format (e.g., "Paris is the capital of [MASK].") to query knowledge in zero-shot settings. In our experiments, $l1$ is set to English. Meanwhile, we set $l2$ to all other 52 languages to report an overall result and rendered deep analysis for 2 similar $l2$ languages (De, Id) and 2 dissimilar $l2$ languages (Ar, Ta)[3]. Moreover, for the overall result, we also

---

[3]While Id does not belong to the same language family as En, it has many similarities with En (Krause, n.d.). Ar and Ta are not considered as Indo-European languages and also do not use latin scripts.

categorize these $l2$ languages into two separate categories for each of the three factors (geographics, writing scripts, and language family) using ISO-639 language code information from "localizely"[4].

**Models.** We examine distinct language model families: xlm-r (0.3B to 10B) (Conneau et al., 2020) and mT0 (0.6B to 3.7B) (Muennighoff et al., 2023). Decoder-based language models are excluded in our study to limit inherent hallucination problems affecting the analysis (Xu et al., 2024; Ji et al., 2023; Fu et al., 2023). In our experiments, we obtain similar findings from both families. Therefore, we only show mT0 results in the main text and move the rest to the Appendix §A.3.

# 3 OBSERVING CONSISTENCY

## 3.1 MAIN FINDINGS

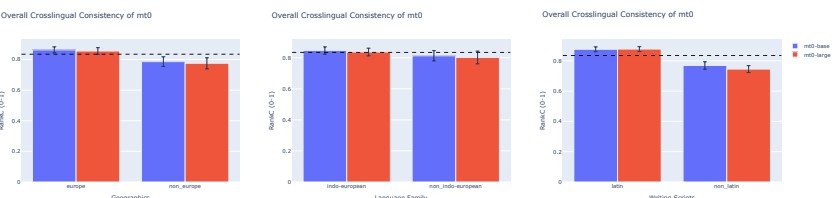

Figure 2: Overall cross-lingual consistency in mt0 (red: mt0-large, blue: mt0-base) grouped by 3 factors (left: geographics, mid: language family, right: writing scripts). Note: The dashed line here is the average corresponding consistency scores of mt0-base across languages( *cf.* §A.3.2. )

From Figure 2, across all the factors, $l2$ that are dissimilar with $l1$, tend to have lower consistency than those are similar to $l1$. The difference in writing scripts plays the most important role among the other two factors. Thus, the number of shared tokens between two languages could affect the cross-lingual consistency, and that is orthogonal to the common view of cross-lingual transfer that shared tokens are not necessary (K et al., 2020; Artetxe et al., 2020). Another intriguing finding is that geographic factor also affects consistency and this could be attributed to common culture and vocabulary (Zhao et al., 2024a). On the other hand, we suppose that other linguistic factors contributing to the cross-lingual performance (de Vries et al., 2022; Kann et al., 2017; Chronopoulou et al., 2023) such as the similarity in linguistic features (Chronopoulou et al., 2023), or borrowing (Tsvetkov & Dyer, 2016), could affect cross-lingual knowledge consistency as well. However, for this study, it is hard to quantify such factors and leave such analyses for future work. Furthermore, we also observe similar results on other models, and this aligns with empirical studies in the literature (Qi et al., 2023). Note that language families and writing scripts have an impact on vocabulary, and we will discuss this vocabulary problem in a later section.

To better understand the cross-lingual consistency bottleneck, we examine the layer-wise consistency patterns across different model sizes, as presented in Figure 3. The noticeable difference lies in the initial consistency, whereby dissimilar language pairs have low consistency scores. The consistency gap between dissimilar and similar languages starts to close at some specific layer while widening again later. This observation provides evidence for empirical studies that scaling benefits the downstream task performance (Conneau et al., 2020), e.g., XNLI, but not be substantially helpful

---

[4]`https://localizely.com/language-code`

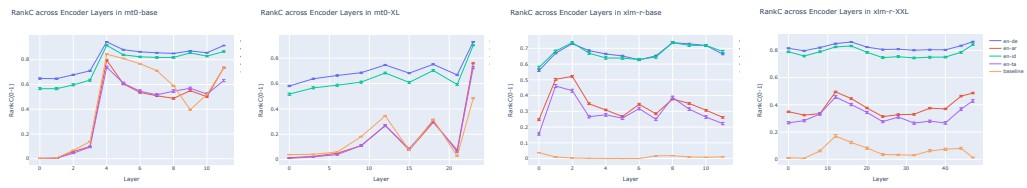

Figure 3: Layer-wise consistency scores in models with different sizes. Scaling models is not a promising strategy to mitigate consistency bottlenecks. ( *cf.* §A.3.1 )

in refining cross-lingual consistency due to cross-lingual consistency bottlenecks. Moreover on Figure 3, we can see the consistency scores of dissimilar languages are quite similar with baseline thus, the consistency of such code-mixed languages are terrible and needed to be improved. Nonetheless, these dissimilar languages are more consistent than our baseline on xlm-r models ( *cf.* 18).

## 3.2 ATTENTION WEIGHT ANALYSIS

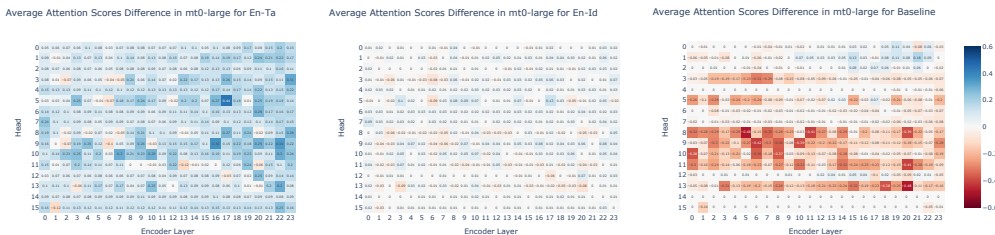

Figure 4: Subject–Object attention difference with $I_{mono}$ to $I_{cm}$ in mt0-large. Attention scores are inversely proportional to the similarity of $l1$ and $l2$ (En–Ta vs En–Id). ( *cf.* §A.3.3)

The layer-wise analyses help us understand the model behaviors. However, the question remains as to how model components handle statements. The correlation analysis conducted on each layer in Table 1 shows that there is a moderate correlation between the average negative subject–object attention scores and the consistency metrics. In particular, although $I_{mono}$ and $I_{cm}$ are coreferential, $I_{cm}$ has to retrieve the reference via the cross-lingual entry. To identify the difference between the cross-lingual and monolingual entries, we observe the attention scores for subject–object pairs. Figure 4 demonstrates that the attention scores across layers and heads are barely distinguishable in the similar language pair, en–id, but more discernable for the dissimilar language pair, en–ta. Sur-

Table 1: Statistical spearman $\rho$ correlation ($\alpha = 0.05$) between average scores of layers with the patterns on each language model's subject-object attention and $IG^2$ absolute difference.

|  | attention | | $IG^2$ | |
| --- | --- | --- | --- | --- |
| Model | RankC | Acc | RankC | Acc |
| mT0-base | 0.414* | 0.426* | 0.528* | 0.519* |
| mT0-large | 0.666* | 0.661* | 0.705* | 0.699* |
| xlm-r-base | 0.433* | 0.424* | 0.400* | 0.397* |
| xlm-r-large | 0.671* | 0.666* | 0.508* | 0.481* |

prisingly, $I_{cm}$ results in higher attention scores than $I_{mono}$ for the dissimilar language pair, which means that the model pays more attention to the subject entity over the predicate across layers and heads in that case. Despite so, we observe from the right side of Figure 4 that having subject–object attention that is too small/big might cause the model to be inconsistent. These insights might be generalizable to different models and other language pairs, as evidenced by Appendix §A.3.3.

## 3.3 $IG^2$ SCORE ANALYSIS

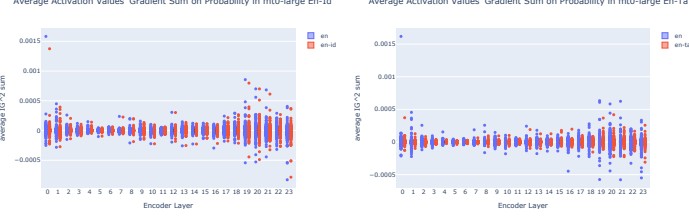

Figure 5: $IG^2$ scores in mt0-large for en–de and en–ta. We see that the distribution is more contrastive on dissimilar languages (En–Ta) than the similar languages (En–ID). (*cf.* §A.3.4. )

In addition, we inspect the $IG^2$ scores of all feed-forward neurons across all encoder layers. Our correlation analysis for this factor could show a moderate correlation with the cross-lingual consistency, as shown in Table 1. In Figure 5, the $IG^2$ scores for similar language pairs are almost the same, while there is a subtle difference for the dissimilar language pairs.

# 4 IMPROVING CONSISTENCY

## 4.1 CAN BIAS CALIBRATE CONSISTENCY?

From previous findings, we think of one question: *can we add biases from $I_{mono}$ to attention layers and feed-forward layers for consistency calibration?* Thus, based on two different patterns discovered from our experiments and having that both are moderately correlated with the consistency score, we do three different causal interventions to align the output of $I_{cm}$ closer to the output of $I_{mono}$. This experiment measures whether each pattern has a causal relationship with cross-lingual consistency. The experimental setup can be seen in Appendix §A.4.1. We consider: **Attention suppression:** suppressing $I_{cm}$'s attention scores to make them closer to $I_{mono}$'s, **Feed-forward neuron activation patching** (Vig et al., 2020; Geiger et al., 2021): patching $I_{mono}$'s activations of all tokens to $I_{cm}$ in selected feed-forward neurons based on $IG^2$, and **Hybrid**: using the above methods simultaneously.

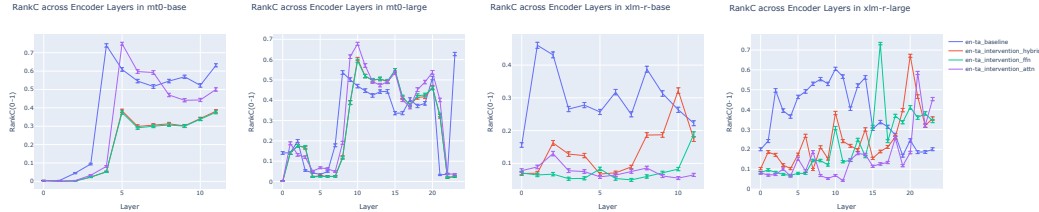

Figure 6: Intervention scores for En–Ta. ( *cf.* §A.4.1).

Based on Figure 6, there is a causal relationship between the two studied factors to some certain extent. For mT0, intervention approaches can increase the consistency scores in the middle-later layers only. While for xlm-r, none of the interventions manages to improve the consistency for the base model despite there is a rising trend for the hybrid intervention and feed-forward activation patching. However, when we observe the larger model, all intervention methods increase the consistency score on the last layers. Overall, our monolingual bias incorporation offers quite limited improvement subject to the architecture and model size thus a better monolingual bias should be considered. These findings are consistent with those of other languages.

## 4.2 THE EFFECT OF VOCABULARY EXPANSION TO THE CROSS-LINGUAL CONSISTENCY

We hypothesize that the size of vocabulary plays a crucial role in improving consistency as this enables a language model to align the semantics better due to lesser ambiguity[5]. To test this hypothesis, we consider two similar language models, xlm-r-base and xlm-v-base (Liang et al., 2023), where xlm-v-base has larger vocabularies (901,629 tokens) than xlm-r-base (250,002 tokens).

---

[5]e.g., if the tokenizer of a language model tokenizes the word "Tokyo" to ["To," "Kyo"], the token "To" is polysemous making thus the alignment of this word would be one-to-many, on the other hand, if a tokenizer keeps the word as it is, the tokenized form of the word is monosemous making it less ambiguous.

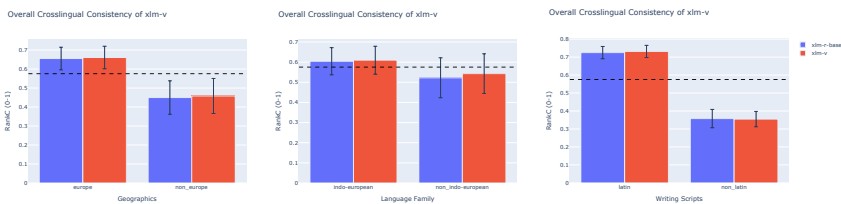

Figure 7: Effects of vocabulary expansion to cross-lingual consistency (red: xlm-v, blue: xlm-r-base). (*cf.* §A.4.2)

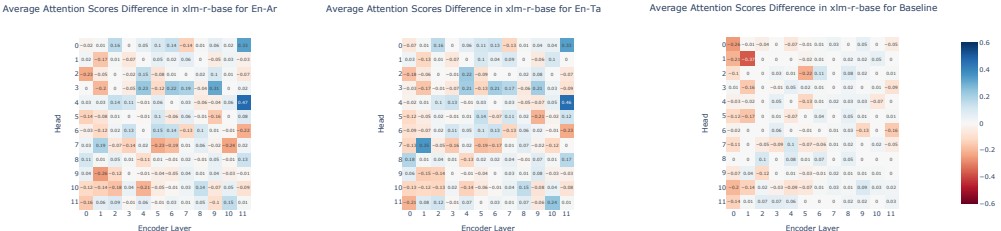

Figure 8: Effects of vocabulary expansion to subject–object attention scores shift to xlm-r-base where we can see there is a shift from earlier layers to middle & last layers. We do not see any notable shift on the baseline input and this is possibly the reason why this method does not offer improvement on the baseline input. (*cf.* §A.4.2)

Vocabulary expansion offers slight consistency improvement for similar and dissimilar language on any categorization, as presented in Figure 7. A large vocabulary limits sub-tokens to prevent the model from latching onto shallow local signals or restoring words from sub-tokens (Levine et al., 2021), which benefits deep semantic learning. However, more samples are required to generalize training. Therefore, vocabulary expansion alone cannot improve the consistency significantly, especially for low-resource languages, but it sitll benefits dissimilar languages with lower consistency in the last layers to alleviate the consistency bottleneck to some extent. This can be observed in Figure 9 that the layer-wise consistency drops significantly in the base model's last layers but increases in the expanded model's last layers. Meanwhile, it can be further confirmed by a similar shift

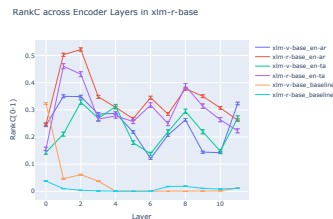

Figure 9: Effects of vocabulary expansion to cross-lingual consistency. (*cf.* §A.4.2)

phenomenon in attention analysis, as shown in Figure 8. The attention scores on the expanded model's last layer or deep semantic layers are more potent than the base model. Since dissimilar language pairs have minimum language features shared across languages, better-aligned semantics alone is not enough to completely resolve the consistency bottleneck. Nonetheless, it is still a crucial aspect to consider for the cross-lingual consistency improvement as demonstrated on the minor improvement shown on Figure 7. It is also in line with Zhao et al. (2024a) where they found that the one-token P@1 of Afrikaans is higher than the Japanese due to segmentation and tokenziation. [6]

## 4.3    THE EFFECT OF CROSS-LINGUAL SUPERVISION TO THE CROSS-LINGUAL CONSISTENCY

Another possible hypothesis is that there might be an entanglement of features between linguistic and knowledge features. El-hage et al. (2022) discovered that the language model (in particular GPT-2 (Radford et al., 2019)) could fit multiple features into one dimension at the price of more entangled features, and this entanglement might cause tokens not cross-lingually aligned as there may be an entanglement between syntactic and semantic features within one dimension. Inspired by that, we suspect this might hinder the consistency of language models. To test this assumption, we evaluate two similar language models in which one model is trained solely on MLM objective (xlm-r), and another similar model is trained on one additional objective to align word translations (xlm-align (Chi et al., 2021)), where this word alignment might be helpful to align references across languages.

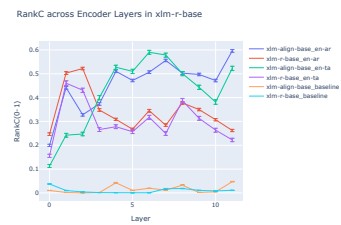

Figure 10: Effects of cross-lingual supervision on the layer-wise consistency. (*cf.* 4.3)

Word alignment increases cross-lingual consistency monotonically to alleviate the cross-lingual bottleneck. Similar to the vocabulary expansion, this strategy does not improve the consistency for the

---

[6]We define this as a token parity issue. See more discussion in Figure 34).

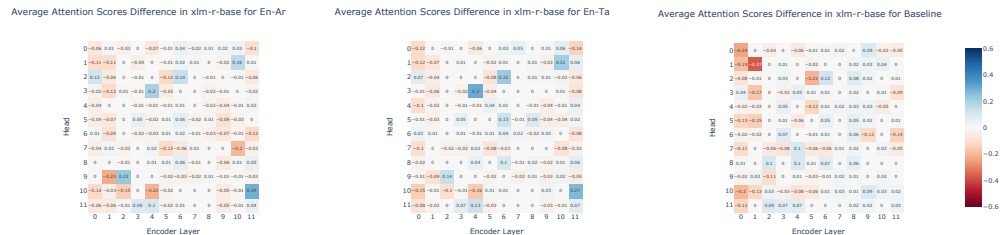

Figure 11: Effects of cross-lingual supervision on subject–object attention in xlm-r-base. We observe a slight layer shift. A similar pattern is observed as the attention difference caused by the vocabulary expansion on the baseline input. left: En–Ar, mid: En–Ta, right: baseline. (*cf.* 4.3)

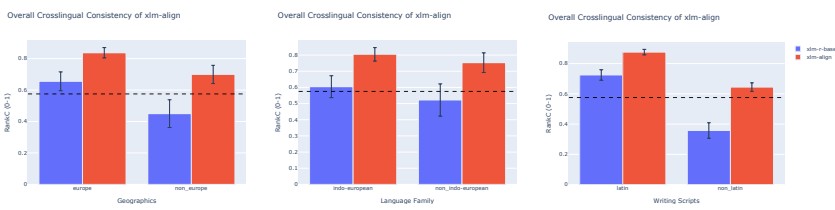

Figure 12: Effects of cross-lingual supervision to xlm-r-base consistency red: xlm-align and blue: xlm-r-base. Note: The dashed line here is the average corresponding consistency scores of xlm-r-base across languages. (*cf.* §A.4.3).

baseline as we would expect. The aligned model outperforms the baseline starting from the middle layers in Figure 10. Multiple pre-training objectives that could approximately disentangle different features can help preserve the model's knowledge of different languages. We could also confirm this finding by observing the overall cross-lingual consistency result in Figure 12, as consistency scores jump over the baseline model's cross-lingual consistency. When we look closer at the object-subject attention scores of the aligned model in Figure 11, there is a slight shift of subject–object relation features extraction from earlier layers (in the baseline model) into middle-last layers (in the aligned model). In line with the vocabulary expansion, the responsibility shift on feature extraction from the earlier layer into later layers might justify the effectiveness of both approaches on dissimilar languages. Interestingly, the attention shift is not as strong as the one caused by vocabulary expansion, which is quite counterintuitive as cross-lingual supervision outperforms vocabulary expansion in improving cross-lingual consistency. Hence, we leave this interesting finding analysis for future work. In addition, word alignments improve consistency for transliterations or similar orthographical forms, contributing to model's robustness against orthographic variations and non-standard spellings, but vocabulary expansion can not offer such gains.(*cf.* §A.4.4)

### 4.4 THE EFFECT OF CODE-SWITCHING TRAINING TO THE CROSS-LINGUAL CONSISTENCY

Inspired by the experiment on cross-lingual supervision, we further hypothesize that code-switching training, which substitutes an entity with alternatives from other languages for intra-sentential alignments in cross-lingual settings, can help the model understand common knowledge across languages for cross-lingual consistency to some extent. To evaluate this hypothesis, we study xlm-r and xlm-r-cs (Whitehouse et al., 2022), where xlm-r-cs is continuously trained on code-switching corpus from xlm-r-base and shows high performance in multilingual fact-checking. From Figure 13, we observe a shift in the consistency bottleneck from the middle layers to the later layers of xlm-r-cs, where the consistency gap between dissimilar and similar languages narrows in xlm-r-cs compared to xlm-r in the middle layers. When observing the attention in Figure 14, we

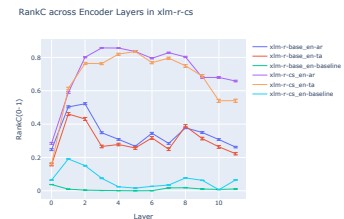

Figure 13: Effects of code-switching training on the layer-wise consistency. (*cf.* 4.3)

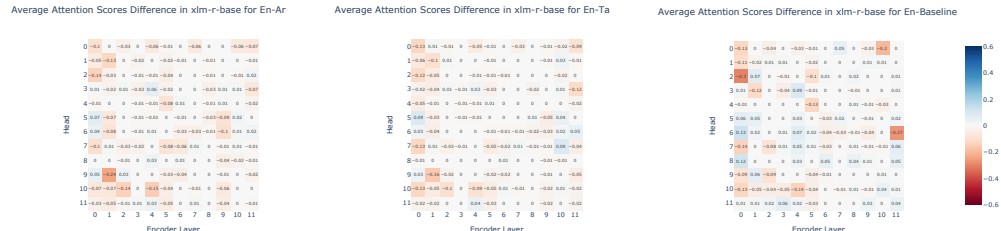

Figure 14: Effects of code-switching training on subject–object attention in xlm-r-base. A similar pattern is observed as the attention difference caused by the vocabulary expansion and the cross-lingual supervision on the baseline input. left: En–Ar, mid: En–Ta, right: baseline. (*cf.* 4.3)

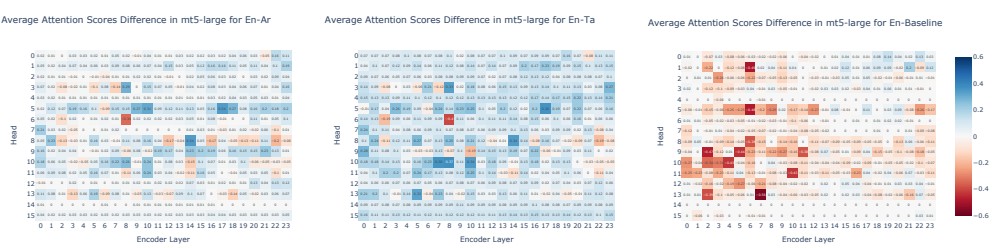

Figure 16: Effects of multi-task fine-tuning on subject–object attention in mt5. We observe a significant head shift but no cross-lingual consistency gains. This is a distinguishable findings from other experiments, where layer behaviors change to mitigate the consistency bottleneck. left: En–Ar, mid: En–Ta, right: baseline. (*cf.* 4.3)

can see one similar layer shift pattern with experiments on the vocabulary expansion and the cross-lingual supervision where the attention weights are suppressed on earlier layers. However, unlike these two approaches, there is no amplification of the attention weights on later layers thus we posit that the key of the improvement probably lies on reducing the responsibility on earlier layers. Therefore, code-switching can offer significant gains to the cross-lingual consistency, even without additional objectives. Overall, this finding is consistent with previous experiments.

## 4.5 THE EFFECT OF MULTI-TASK FINE-TUNING TO THE CROSS-LINGUAL CONSISTENCY

In previous discussions, we discussed the cross-lingual consistency in the mt0 family, which is multi-task fine-tuned from the mt5 family (Xue et al., 2020). We hypothesize that this fine-tuning can improve the cross-lingual consistency due to improved cross-lingual generalization across similar tasks in different languages, as opposed to word-level alignments discussed in previous sections. Surprisingly, multi-task fine-tuning can not offer significant gains to the cross-lingual consistency. As presented in Figure 15, the consistency patterns are quite similar across mt0 and mt5. Instead of shifting layers for the attention, which is observed in other experiments, multi-task fine-tuning causes a head shift in Figure 16. We suspect that the model adjusts some neurons at each layer to maintain knowledge but such

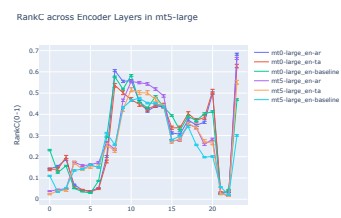

Figure 15: Effects of multi-task fine-tuning on the layer-wise consistency. (*cf.* 4.3)

adjustment has limited contributions to cross-lingual consistency. While this finding is distinguishable from most of other methods, it is consistent with Figure 4, where we found scaling is not promising to improve cross-lingual consistency. Specifically, both of them encourage some neurons to preserve the cross-lingual knowledge consistency, showing limited effectiveness. This finding aligns with Ortu et al. (2024); Jin et al. (2024) who reported that LLM has attention heads with contrasting roles in which some of them consider retrieving internal knowledge of language models and other heads prefer to get the in-context information.

## 5 RELATED WORK

Petroni et al. (2019) found the availability of relational knowledge within the pre-trained language model by evaluating the language models on the cloze task dataset they proposed, namely LAMA. Kassner et al. (2021) introduced the multilingual version of it, mLAMA, and discovered that the language's relational knowledge capability varies in different languages and other works also found similar findings (Schott et al., 2023; Zhao et al., 2024a). Nevertheless, Zhao et al. (2024a) showed that multilingual language models exhibited limited cross-lingual knowledge recall capability on low-resource languages. Following this line, Fierro & Søgaard (2022); Qi et al. (2023) studied the final predictions in different languages and reported inconsistencies across languages. Moreover, Jin et al. (2024) proposed a method to mitigate such conflicting mechanisms by nullifying heads having significant impact in either of both roles. We take a different angle from those works where we evaluate the cross-lingual knowledge consistency against references in different languages by creating coreferential statements in cross-lingual settings.

Bhattacharya & Bojar (2023); Kojima et al. (2024); Zhao et al. (2024b) discovered the language-sensitive neurons of decoder in the early and last layers while a considerable portion of language-agnostic ones in the middle layers encode universal concepts and utilize the latent language (in this case English) (Wendler et al., 2024; Dumas et al., 2024). Tan et al. (2024) observed encoder-decoder language models that neurons tend to be more language-agnostic in the later layers of the encoder part while language-specific in the later layers of the decoder part. Zhao et al. (2024b); Wang et al. (2024b); Zhang et al. (2024) further showed the cross-lingual downstream performance is potentially proportional to the amount of language-agnostic neurons. Ferrando & Costa-jussà (2024) discovered a shared circuit or subnetwork that is responsible for subject-verb agreement task for English & Spanish and Stanczak et al. (2022); Wang et al. (2024a) found that morpho-syntax attributes have noticeable neuron overlapping degree over notable amount of language pairs. We push this line further to trace consistent information and knowledge throughout the layers in cross-lingual settings, attempting to understand and interpret how commonly used strategies to improve multilingual models for downstream tasks could impact the cross-lingual knowledge consistency.

## 6 CONCLUSION

Do multilingual language models demonstrate cross-lingual consistency? Is it worthwhile to optimise for cross-lingual knowledge consistency? We find the answer to both of these questions is 'yes', but with the caveat that performance is tied to language characteristics. In our work, we code mix source monolingual sentences containing a coreferential named entity to control and analyze cross-lingual knowledge consistency.

Our analysis reveals that knowledge consistency is heavily dependent on language-specific information such as geography, language family, and writing script. Our layer-by-layer analysis of multilingual models discovers a consistency bottleneck in the middle layers of models. This bottleneck can be alleviated by expanding the vocabulary, injecting cross-lingual supervision and in training, or including code-swithcing corpus. Our work highlights promising directions in post-calibration, vocabulary formation, pretraining with cross-lingual objectives, and code-switching training to achieve knowledge consistency across languages, which will better preserve parity of language model performance. As our experiment discovers that pretraining objective and code-switching training cause most significant positive impact on the cross-lingual consistency, we encourage researchers to emphasize more on representation learning approaches to make the language models more consistent across different languages.

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

## LIMITATIONS

We only cover the transformer encoder and encoder-language models for this work. Another promising avenue for this work is evaluating cross-lingual knowledge consistency on other language models. Moreover, we only analyze each crucial component independently due to the time constraint and left scrutinizing the interaction between each component for future work. In the future, we may expand this work by analyzing how the interaction among these components could affect the cross-lingual consistency of multilingual models. Another thing is that our causal intervention method needs to be done manually, and we suspect that this method could produce a side effect on the model because the representations encoded by language models are more likely to be polysemous. Additionally, we only evaluate the language models in context-independent settings. Thus, in the future, we plan to evaluate the consistency of the models' knowledge and observe whether language models utilize their parametric knowledge more or emphasize the knowledge from the given context under the cross-lingual setting. Another thing to consider is that we only evaluate our solution using one particular model due to the time constraint. Also, we do not explore various pre-training objectives and evaluate solutions to the encoder-decoder model; hence, we leave such things as future work. One interesting thing to explore in this aspect is to see whether adversarial training could help to enhance cross-lingual consistency. Another thing that we want to consider is that we use an assumption that one reference is represented as a single English object entity to make the evaluation tractable; hence, we do not take into account the real-world setting where one reference can be interpreted in different ways on multiple languages (e.g., "China" is written as "ZhongGuo" in Chinese rather than "China"). Lastly, our research scope assumes that the knowledge we want to evaluate is factual and not dependent on subjective aspects (e.g., cultural context). With that assumption, we assume that references here generally have one-to-one mapping to representation in one language where the representation here is considered common knowledge.

## ETHICS STATEMENT

This work aims to evaluate the consistency of the language model across different senses (particularly between a monolingual input and its code-mixed counterparts) and the impact of different factors on that metric. Doing such a study could shed light on the limitations of language models and think of the mitigations of such matters.

### REPRODUCIBILITY STATEMENTS

We used open-source pretrained models and also dataset for all of the reported experiments thus no undisclosed assets utilized in our work. Additionally, we also provide necessary experiments' output and codes on `https://anonymous.4open.science/r/knowledgeConsistencyAndConflict-4827`.

Table 2: Input sample for the evaluation task. We only predict the object in bold. $I_{\backslash(S^{l1}\cap O)}$ is the baseline input.

|  | xlm-r input | mt0 input |
| --- | --- | --- |
| $I_{mono}$ | Paris is the capital of <**mask**> | Paris is the capital of <**extra_id_0**> |
| $I_{cm}$ | باريس is the capital of <**mask**> | باريس is the capital of <**extra_id_0**> |
| $I_{\backslash(S^{l1}\cap O)}$ | <mask> is the capital of <**mask**> | <extra_id_0> is the capital of <**extra_id_1**> |

# A APPENDIX

## A.1 INPUT FORMAT

In our task definition, we introduce our evaluation task in both intuition and math perspective. Here is the input sample in Table 2. Meanwhile, as presented in the task definition, we do not consider whether predictions are true but focus on the same prediction distributions regardless of languages. Note that we did not perturb the surface structure in order to minimize variables to affect factual knowledge recall because $S^{l2}$ "switches-in" at grammatically correct point as the new subject (Pratapa et al., 2018).

## A.2 Metric Function and Interpretability Approach

**RankC** RankC (Qi et al., 2023) is used to evaluate the cross-lingual knowledge consistency. Given a set of statements $S$ where each of the statement having each own $I_{mono}$ and $I_{cm}$, the number of candidates $Cand(O_{\in V}|I_{mono})$ of i-th statement $N_i$, $mono^j$ stands for the j-th candidate of $Cand(O_{\in V}|I_{mono})$, $cm^j$ stands for the j-th candidate of $Cand(O_{\in V}|I_{cm})$, and the RankC score of $Cand(O_{\in V}|I_{mono})$ concerning $Cand(O_{\in V}|I_{cm})$ can be written as

$$RankC(cm, mono) = \frac{\sum_{i=1}^{|S|} \sum_{j=1}^{N_i} \frac{e^{N_i - j}}{\sum_{k=1}^{N_i} e^{k-j}} * P@j}{|I_{mono}|},$$ (1)

$$P@j = \frac{1}{j}|\{cm_i^1, cm_i^2, \cdots, cm_i^j\} \cap \{mono_i^1, mono_i^2, \cdots, mono_i^j\}|.$$ (2)

**Top@1 Accuracy** The Top@1 accuracy is defined as the average number of exact matches between the top-1 predictions given $I_{mono}$ and $I_{cm}$.

**Subject–Object Attention** Let $A_{a,b}^{(k)}$ be the attention score between $a$-th token and $b$-th token in a statement $k$, $O_k$ is the set of indices of the masked tokens in $k$, $S_k$ is the set of indices of subject tokens in $k$, and $K$ is a set of statements, the average attention weight of head $l$ in $i$ layer can be defined as

$$Attn(h^{(l,i)}) = \frac{\sum_{k \in K} \frac{\sum_{o \in O_k} \sum_{s \in S_k} A_{o,s}^{(k)}}{|O_k|}}{|K|}$$ (3)

**$IG^2$ Score** If $w_j^{(l)}$ is the activation value of $j$-th neuron in the $l$-th layer of a particular input (either code-mixed or not), $m$ is the approximation step, and $t$ as a token of the whole ground truth object entity, the score for a given $I_{mono}$ or $I_{cm}$ is defined as

$$IG^2(w_j^{(l)}) = \sum_{t \in T} \frac{\frac{w_j^{(l)}}{m} \sum_{k=1}^{m} \frac{\partial P(t|\frac{k}{m} w_j^{(l)})}{\partial (\frac{k}{m} w_j^{(l)})}}{|T|}$$ (4)

## A.3 FINDINGS IN DETAILS

### A.3.1 LAYER-WISE CONSISTENCY

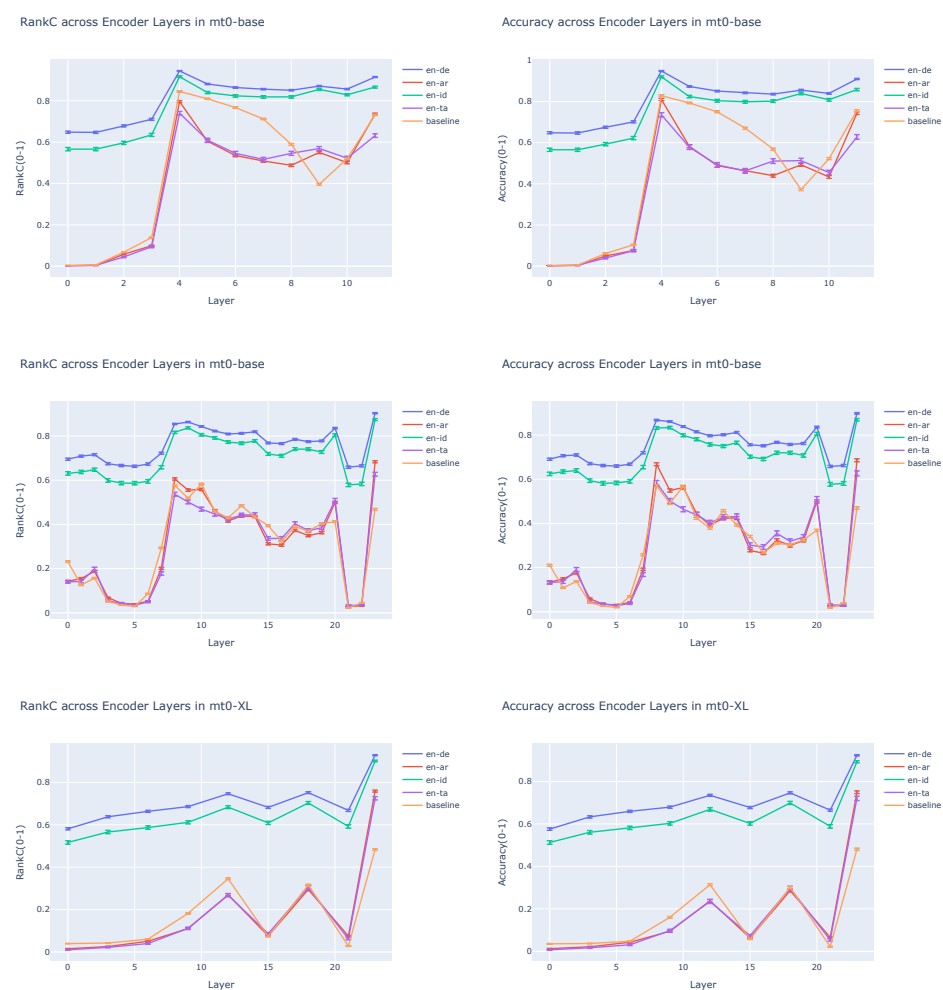

Figure 17: mT0 layer-wise cross-lingual consistency scores (left: RankC, right: Top@1)

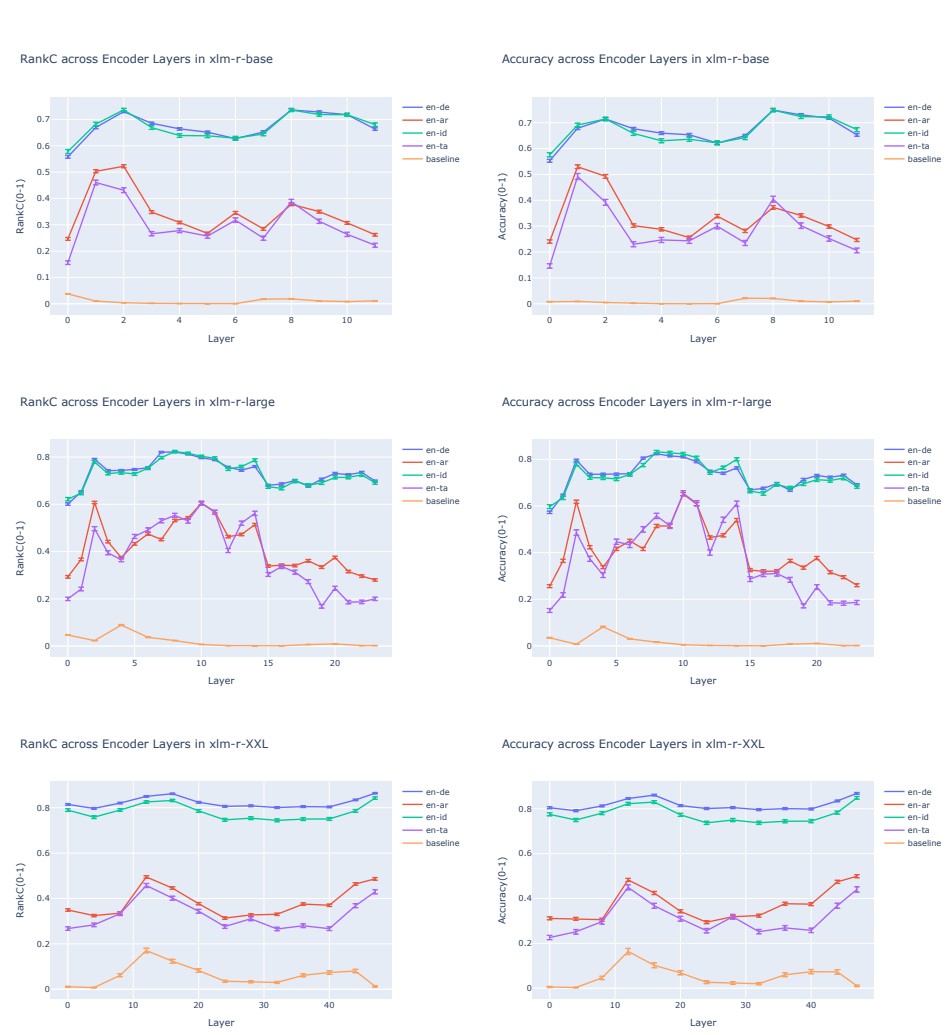

Figure 18: xlm-r layer-wise crosslingual consistency scores (left: RankC, right: Top@1)

A.3.2 OVERALL CONSISTENCY

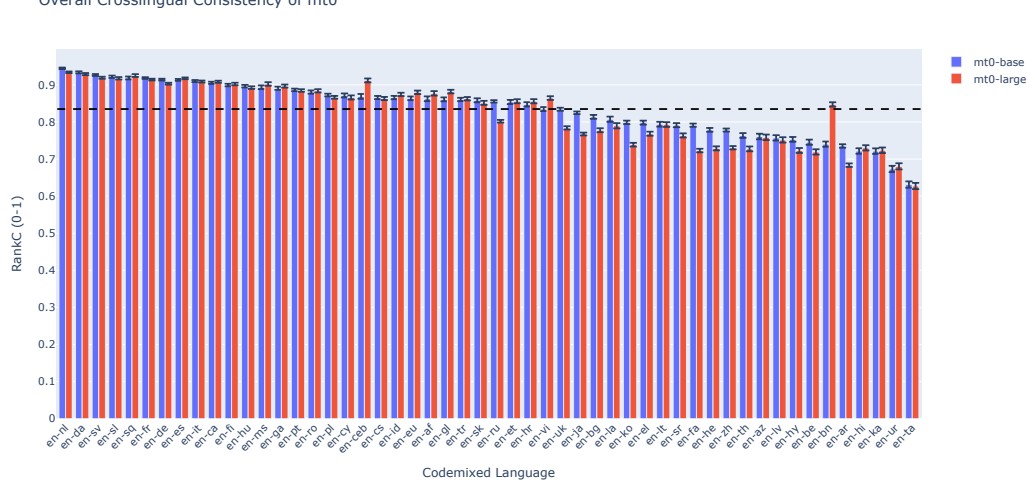

Figure 19: Cross-lingual consistency scores across languages of mt0 (top: RankC, bottom: Top@1 Accuracy). Note: The dashed line here is the average corresponding consistency scores of mt0-base across languages

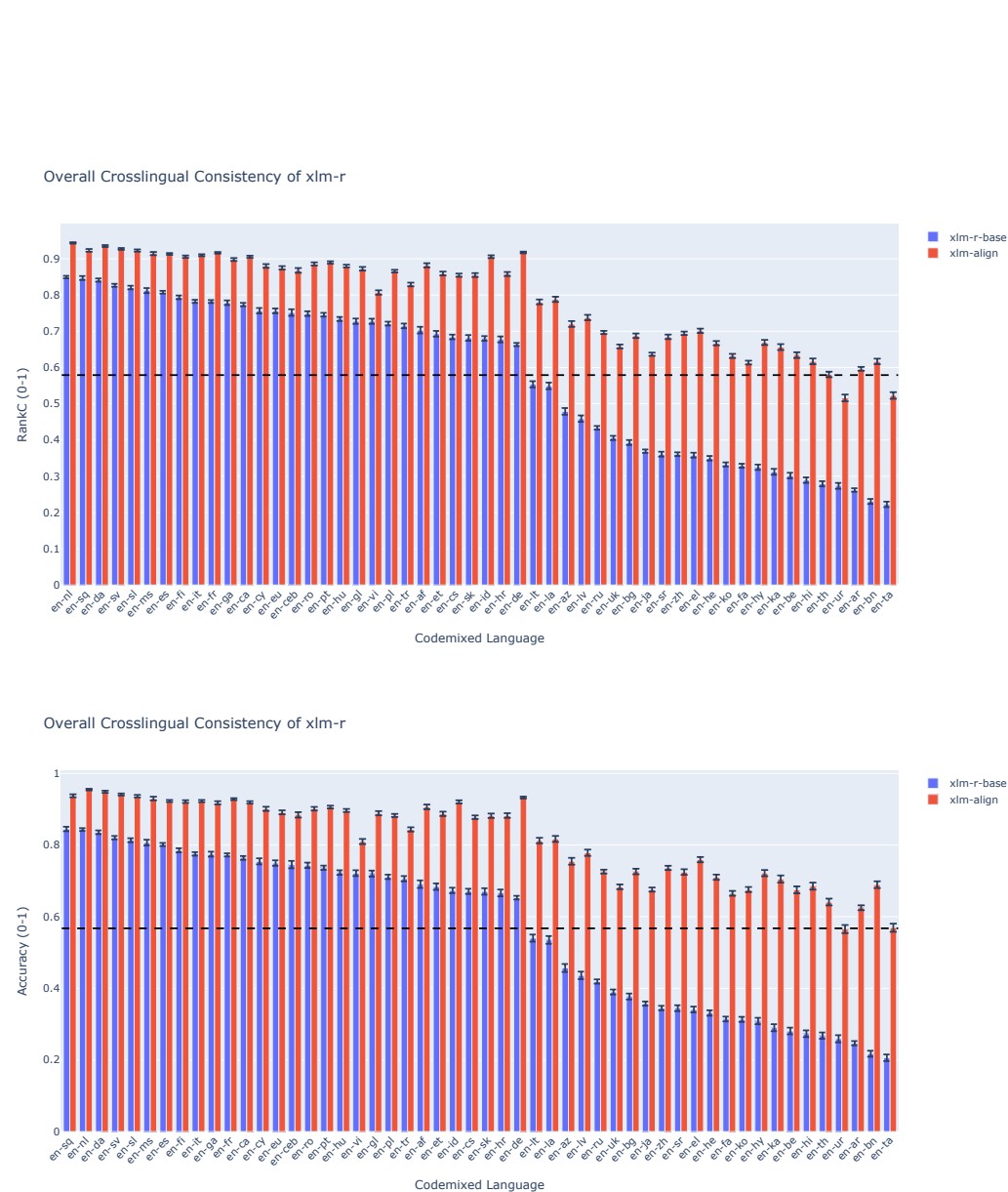

Figure 20: Cross-lingual consistency scores across languages of xlm-r (top: RankC, bottom: Top@1 Accuracy). Note: The dashed line here is the average corresponding consistency scores of xlm-r-base across languages

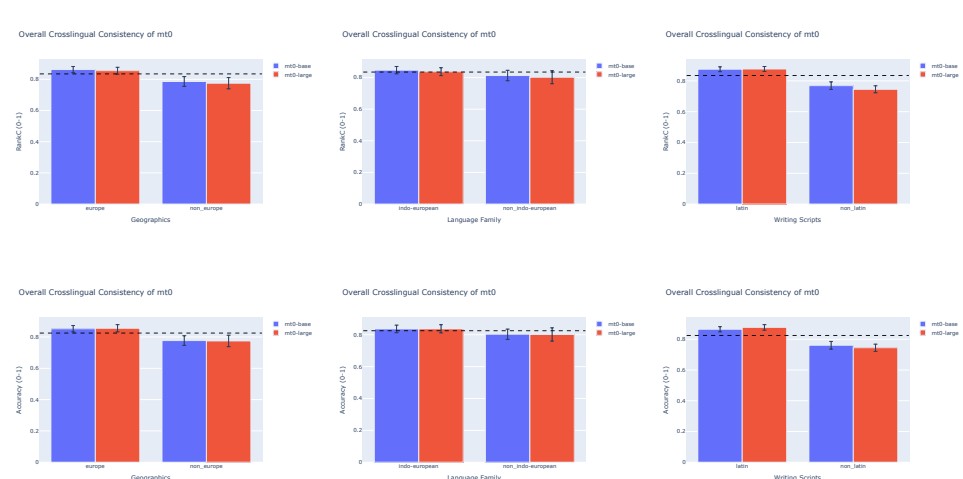

Figure 21: Overall cross-lingual consistency in mt0 grouped by 3 factors (left: geographics, middle: language family, right: writing scripts.). Metrics legend: top: RankC, bottom: Top@1 Accuracy. Models legend: red: mt0-large, blue: mt0-base

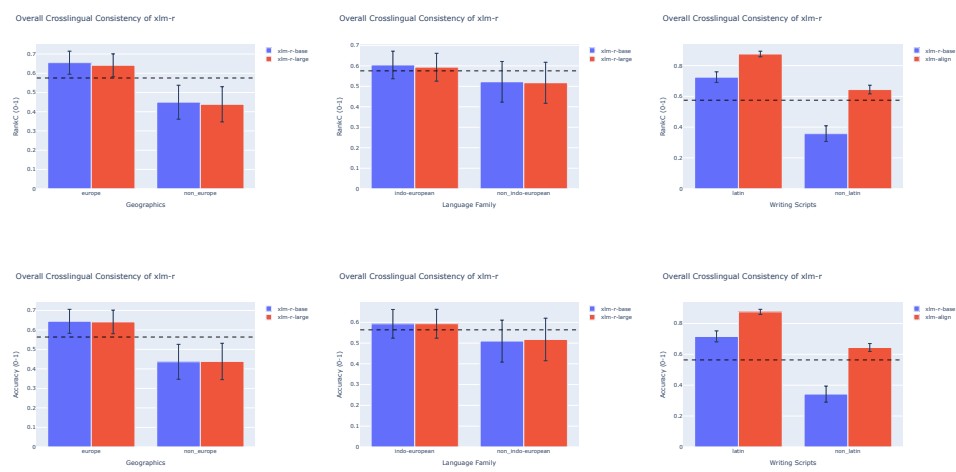

Figure 22: Overall cross-lingual consistency in xlm-r grouped by 3 factors (left: geographics, middle: language family, right: writing scripts). Metrics legend: top: RankC, bottom: Top@1 Accuracy. Models legend: red: xlm-r-large, blue: xlm-r-base

## A.3.3 Subject–Object Attention Score Differences

Figure 23: Subject–Object attention difference with $I_{mono}$ to $I_{cm}$ in mT0 for some code-mixed languages. Models legend: left: mt0-base, right: mt0-large. Languages legend: from top to bottom, en–de, en–ar, en–id, and en–ta.

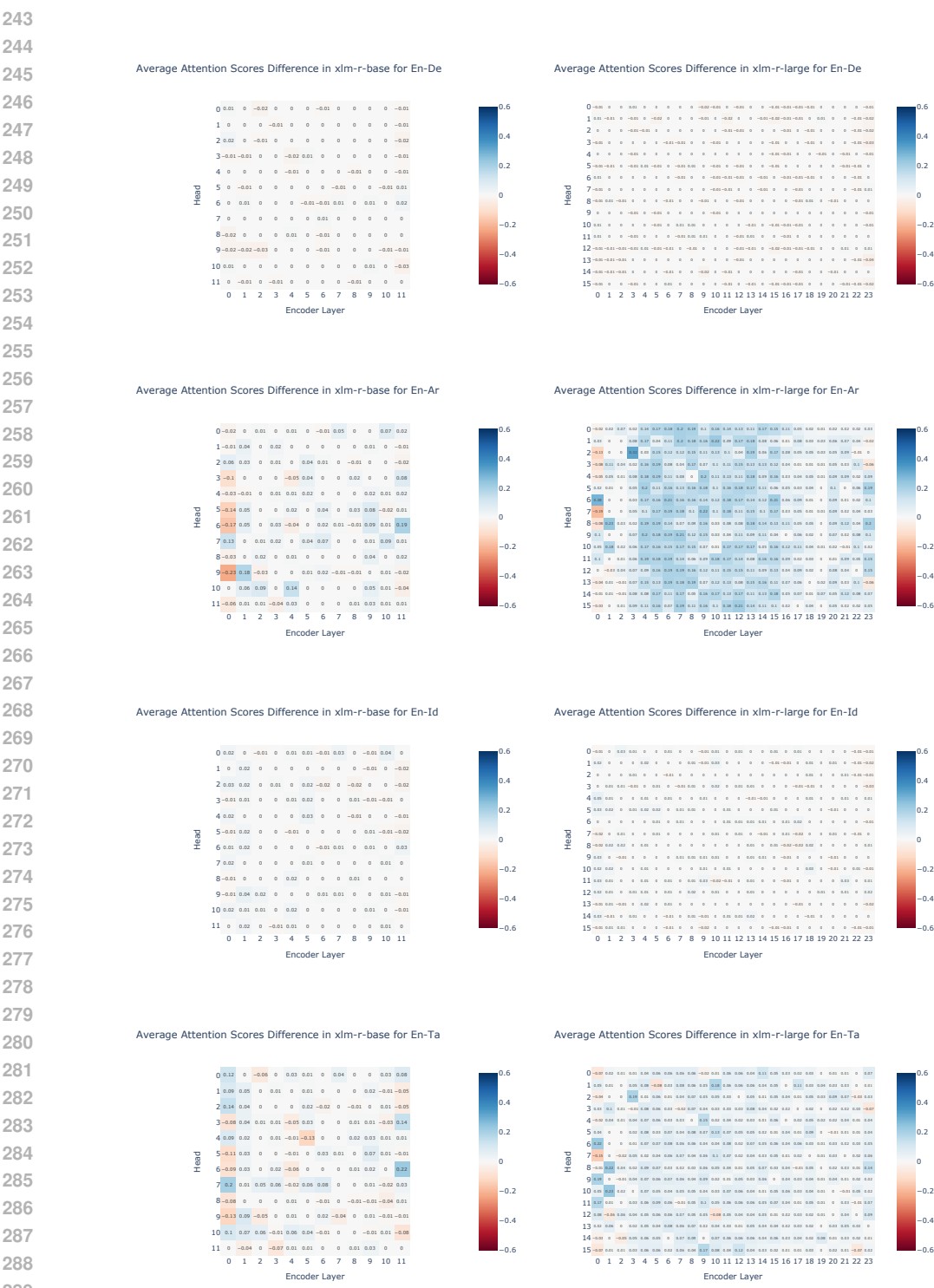

Figure 24: Subject–Object attention difference with $I_{mono}$ to $I_{cm}$ in xlm-r for some code-mixed languages (From left to right, base model and large model. from top to bottom, en–de, en–ar, en–id, and en–ta).

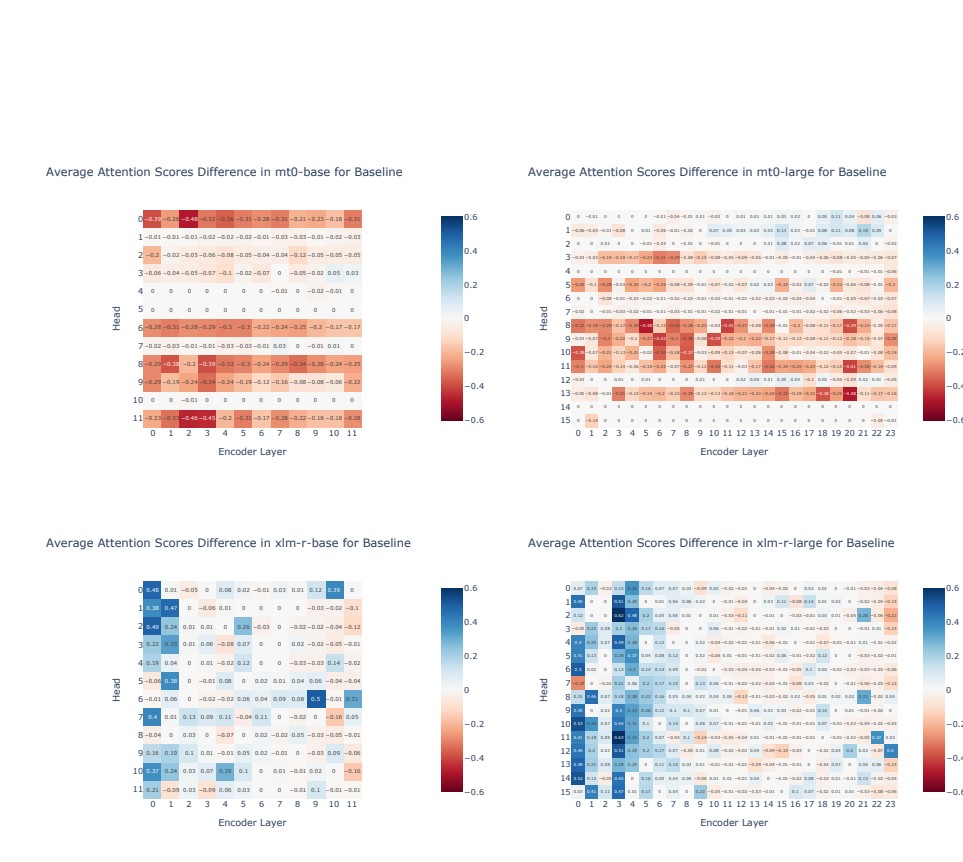

Figure 25: Subject–Object attention difference with $I_{mono}$ to $I_{cm}$ in all models for baseline codemixed input. Models size legend: left: base, right: large. Models family legend: top: mT0, bottom: xlm-r.

### A.3.4 Feed-Forward Neurons' Gradients Sum

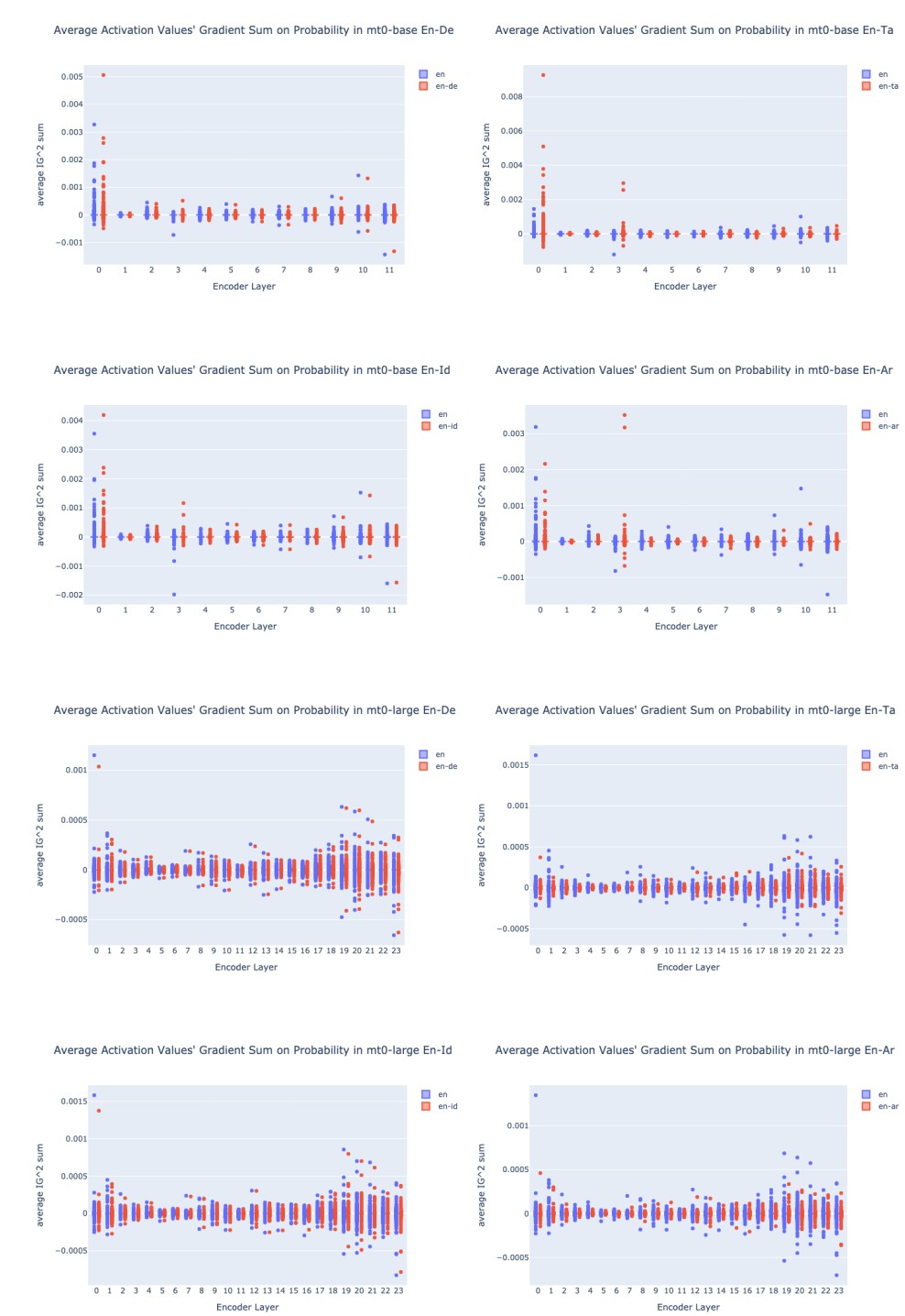

Figure 26: $IG^2$ scores in mt0 for en–de, en–ta, en–id, and en–ar. Models legend: upper two rows: mt0-base, lower two rows: mt0-large.

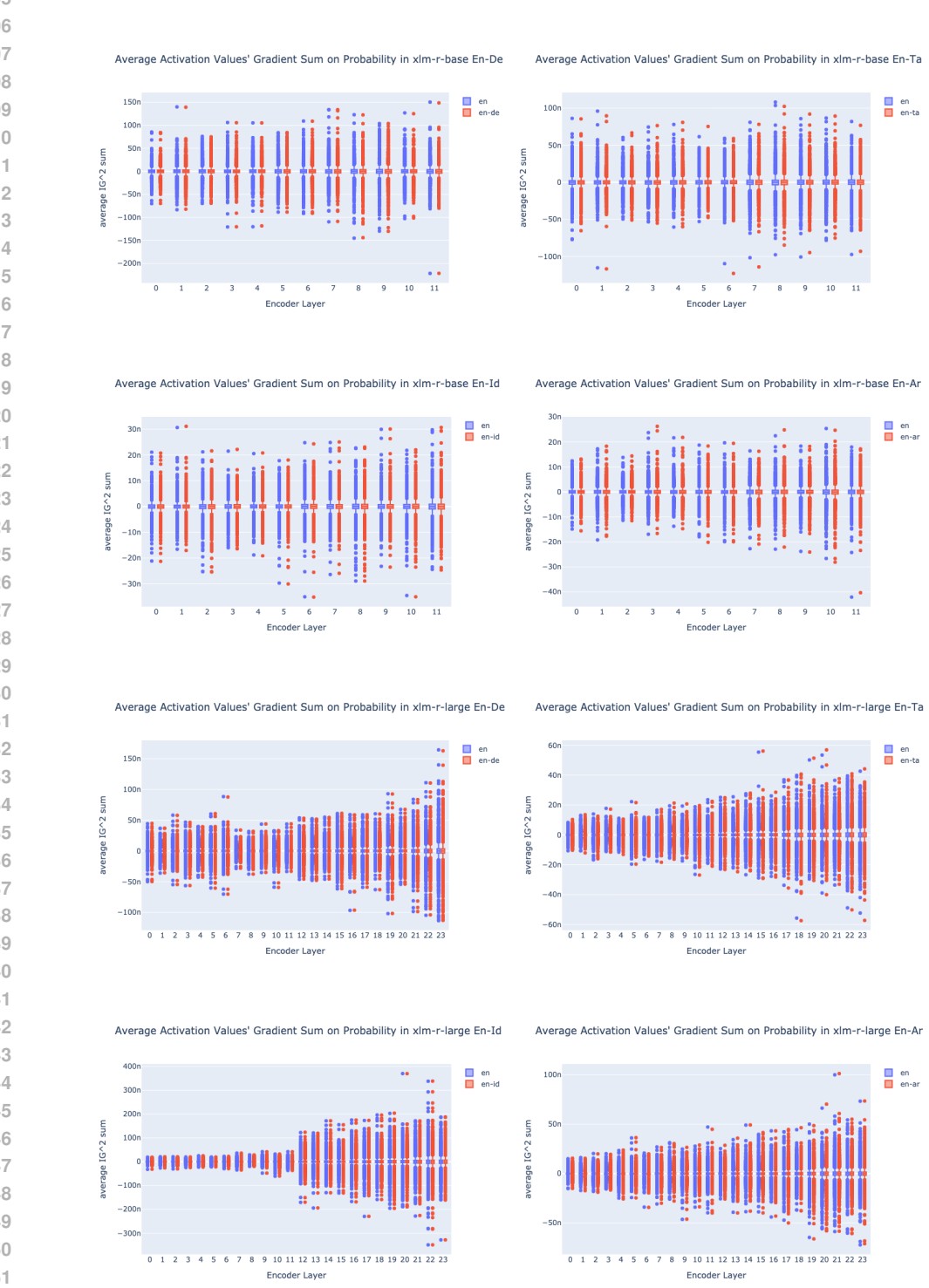

Figure 27: $IG^2$ scores in xlm-r for en–de, en–ta, en–id, and en–ar. Models legend: upper two rows: xlm-r-base, lower two rows: xlm-r-large

| Model | Codemixing Language | $\alpha$ | Patched FFN Layers |
|---|---|---|---|
| mt0-base | en–ta | 0.7 | [0,3,10,11] |
| | en–ar | 0.7 | [0,1,9,10] |
| mt0-large | en–ta | 0.8 | [0,1,19,20,21] |
| | en–ar | 0.8 | [0,1,19,20,21] |
| xlmr-base | en–ta | 0.7 | [5,8,9,10] |
| | en–ar | 0.7 | [5,7,8,10] |
| xlmr-large | en–ta | 0.7 | [0,2,5,19,20] |
| | en–ar | 0.8 | [17,18,19,20,21] |

Table 3: Causal Intervention Hyperparameters Setup

## A.4  IMPROVING CONSISTENCY

### A.4.1  ADDING MONOLINGUAL BIAS

This experiment aims to measure whether each pattern has a causal relationship with cross-lingual consistency.

- Attention score suppression: Using the definition from A.2 and define a suppression constant $\alpha, \alpha \in [0,1)$, the patched attention weight of every object-subject relation will be $A^*_{a,b} = \alpha A_{a,b}$.

- Feed-forward neuron activation patching (Vig et al., 2020; Geiger et al., 2021): consider $a^{(l,p)}_i$ as the activation of $i$-th token on $I_{mono}$ produced by $p$-th neuron in $l$-th encoder layer's feed-forward network, then patched activation value for the $i$-th token on $I_{cm}$ is $\bar{a}^{(l,p)}_i = a^{(l,p)}_i$, in which we apply this for every mask token.

- Hybrid: We apply attention weight suppression and feed-forward neuron activations patching simultaneously.

For the hyperparameters used in the causal intervention experiment, we set the $\alpha$ value that is not too big so that did not significantly diminish the attention weight yet making the attention weight distribution for $I_{cm}$ closer to that weight distribution for $I_{mono}$. While for FFN-layers, we intervene 4 different encoder layers for base models and 5 different encoder layers that have language-sensitive neurons based on $IG^2$ (i.e. layer which has noticeable $IG^2$ distribution difference between $I_{mono}$ and $I_cm$). Readers can refer to table 3 to see the hyperparameters used in this experiment.

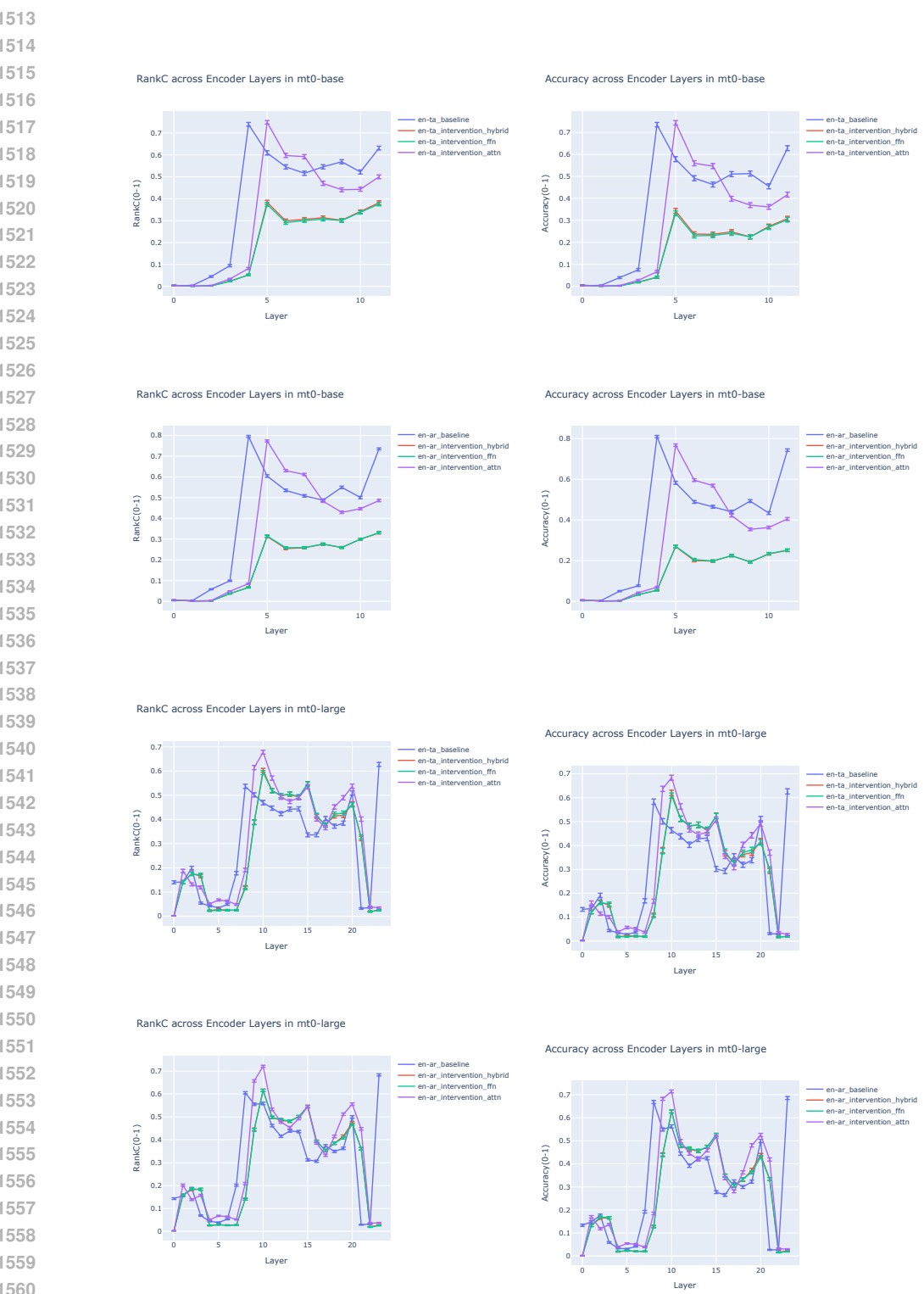

Figure 28: Intervention scores in mt0. Metrics legend: left: RankC, right: Top@1 Accuracy. Models legend: upper two rows: mt0-base, lower two rows: mt0-large

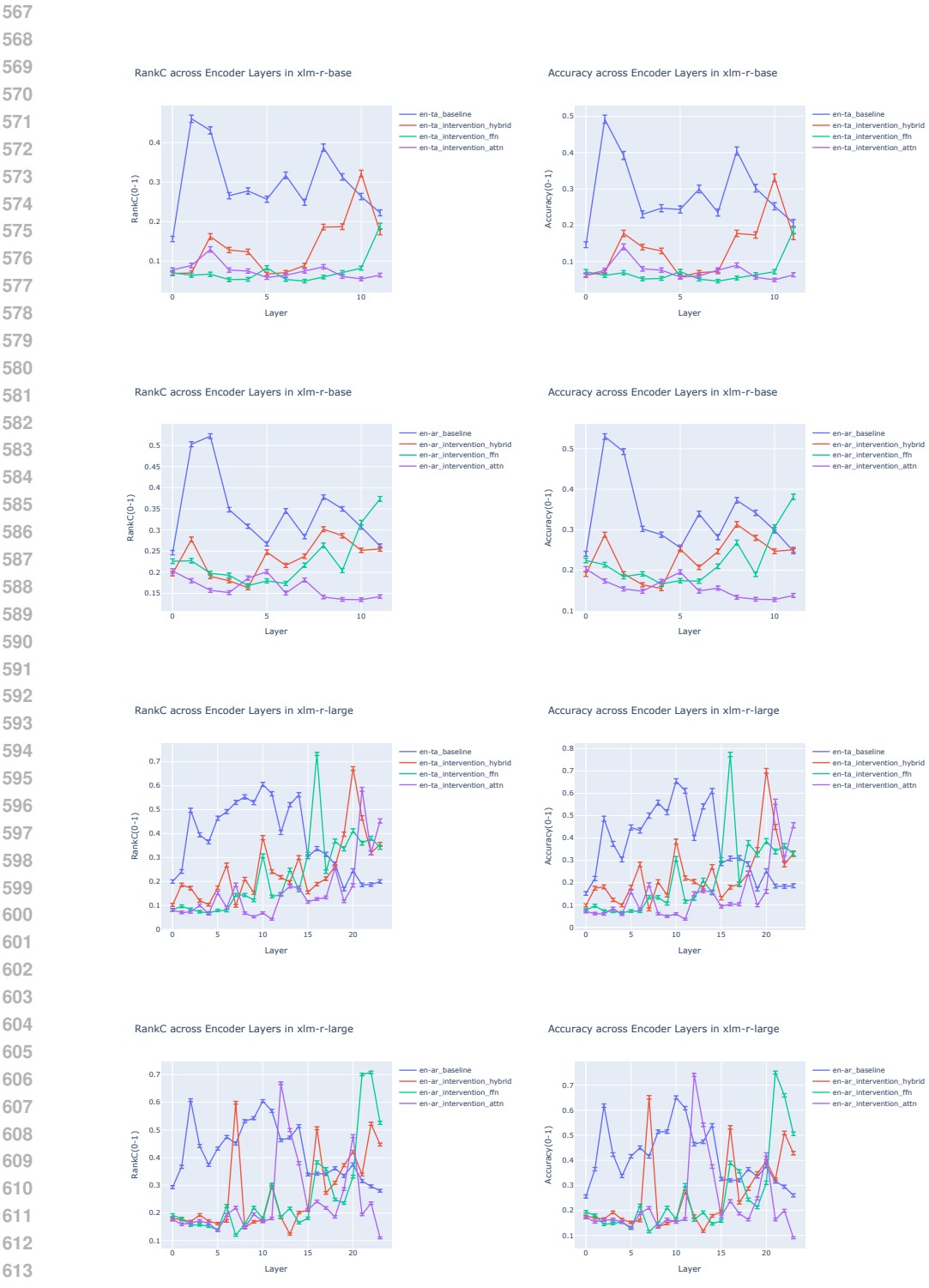

Figure 29: Intervention scores in xlm-r. Metrics legend: left: RankC, right: Top@1 Accuracy. Models legend: upper two rows: xlm-r-base, lower two rows: xlm-r-large

### A.4.2 IMPACT OF LARGER VOCABULARY

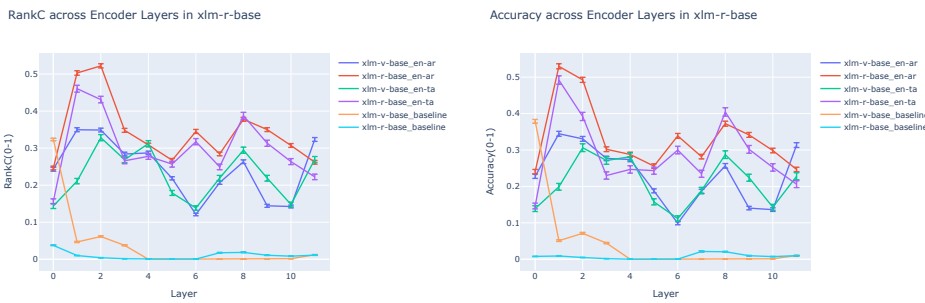

Figure 30: Layer-wise cross-lingual knowledge consistency of xlm-v vs xlm-r-base

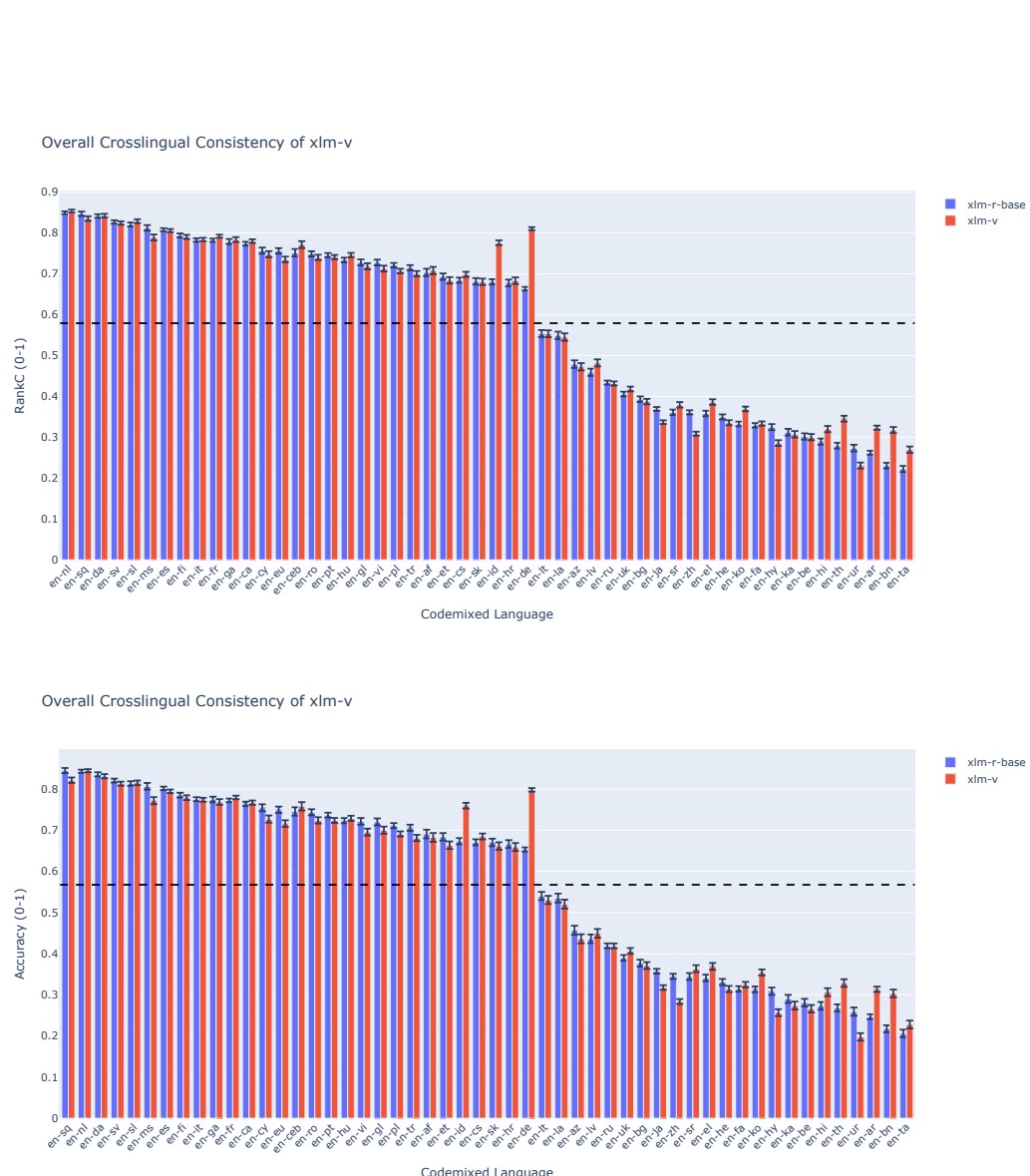

Figure 31: Effects of vocabulary expansion to overall cross-lingual consistency (top: RankC, bottom: Top@1 Accuracy). Note: The dashed line here is the average corresponding consistency scores of xlm-r-base across languages

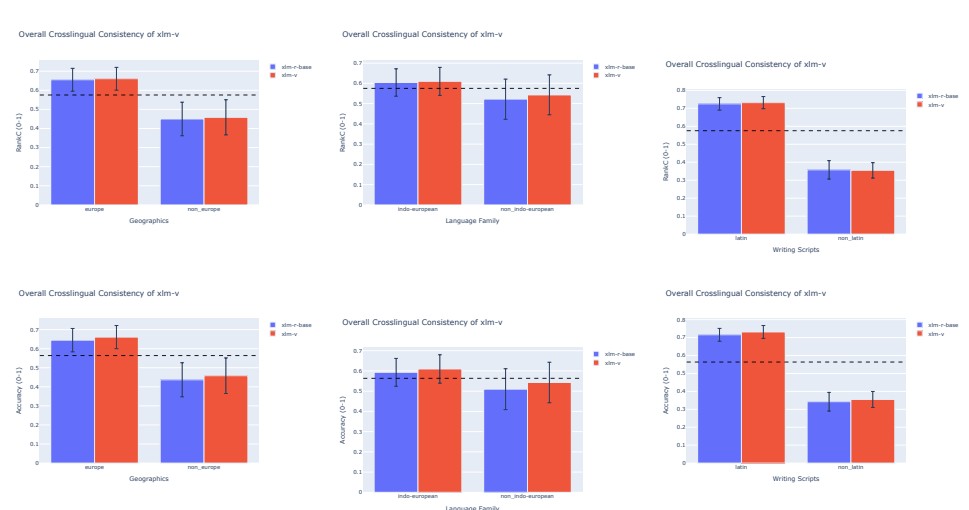

Figure 32: Effects of vocabulary expansion to xlm-r-base (aggregated scores). Metrics legend: top: RankC, bottom: Top@1 Accuracy. Models legend: red: xlm-v, blue: xlm-r-base

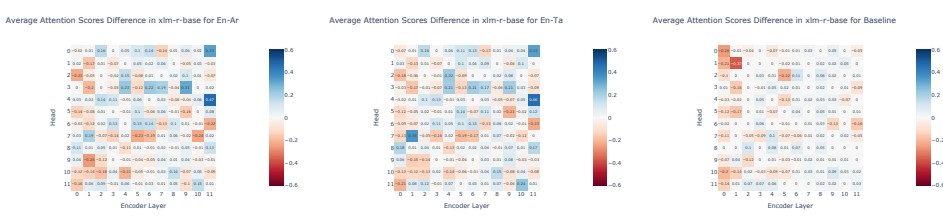

Figure 33: Effects of vocabulary expansion to subject–object attention scores shift to xlm-r-base where we can see there is shift from earlier layers to middle & last layers for dissimilar languages. left: en–ar, mid: en–ta, right: baseline

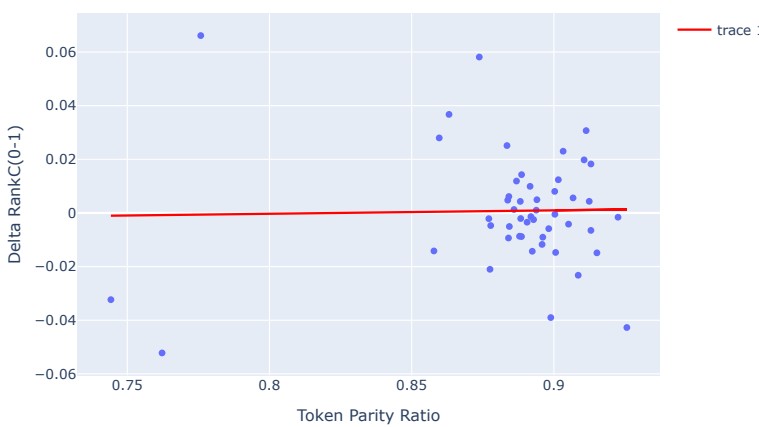

Figure 34: Regression analysis between parity ratio and RankC improvement offered by xlm-v to xlm-r. Spearman $\rho = 0.06$. We define parity ratio as the token length ratio between tokenized subjects for xlm-v-base and xlm-r-base. Our analysis discovers that many languages have a token parity ratio average within 0.8-1, which means that many of the subject entities are known on both tokenizers of the models.

### A.4.3 THE EFFECT OF PRE-TRAINING OBJECTIVE

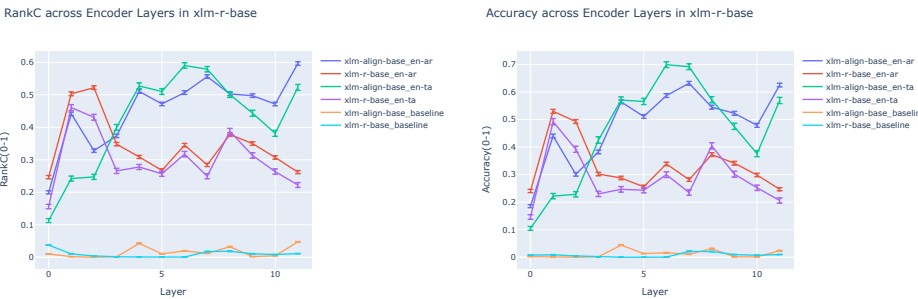

Figure 35: Layer-wise cross-Lingual knowledge consistency of xlm-align vs xlm-r-base

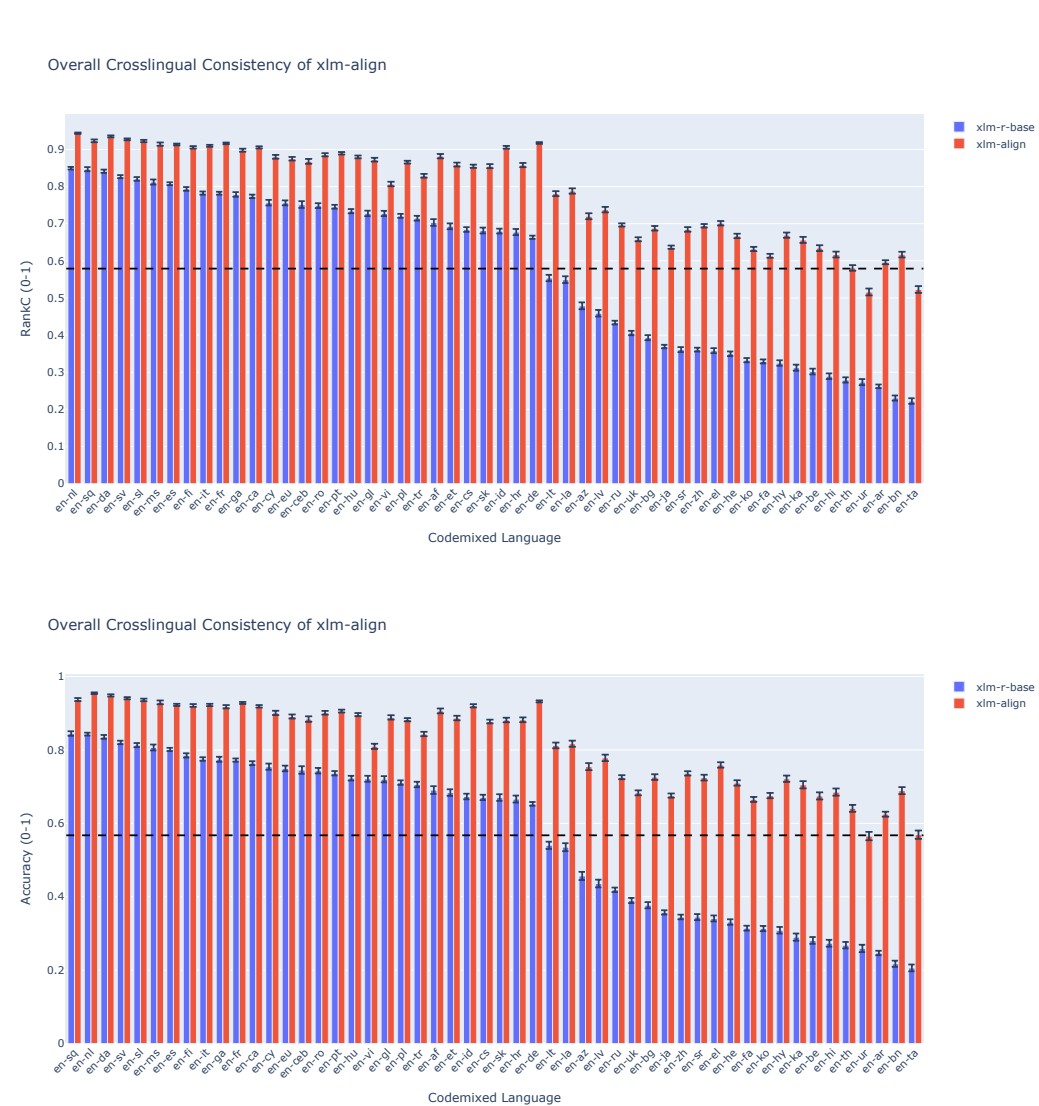

Figure 36: Effects of pretraining objective to overall cross-lingual consistency (top: RankC, bottom: Top@1 Accuracy).

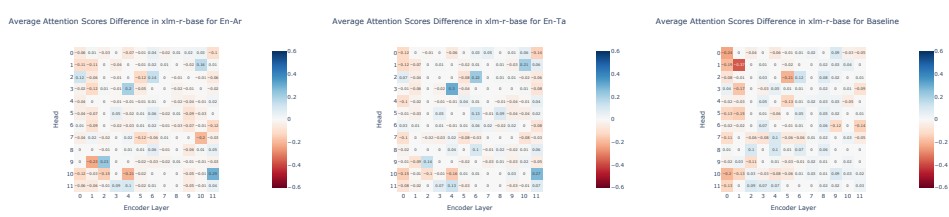

Figure 37: Effects of cross-lingual supervision to subject–object attention scores to xlm-r-base. We can see there is a slight layer shift on the attention for dissimilar languages. left: en–ar, mid: en–ta, right: baseline

### A.4.4 CASE STUDY FOR TRANSLITERATION

Instead of using translations, we transliterate bn[7] and ar[8] to understand the impact of writing systems, particularly transliterations. As presented in Figure 38, word alignments (or the similar effect from CS training) contribute to the model's cross-lingual consistency against writing systems because xlm-align and xlm-r-cs show similar performance in both original and transliteration settings. Meanwhile, we can observe that xlm-align and xlm-r-cs significantly improve the overall performance for non-Lattin scripts in §A.4.3. This is reasonable as word alignments or CS training help the model link original words with their translations or transliterations, depending on the training corpus, thereby enhancing cross-lingual consistency. We suspect that these word alignments might also improve robustness for handling non-standard spellings and orthographic variations. However, xlm-v-base and xlm-r-base without word alignment benefit from transliterations, which means that xlm-v-base and xlm-r-base do not sufficiently align original words with their transliterations to main cross-lingual consistency. It is also confirmed by the overall performance of vocabulary expansions in §A.4.2, where vocabulary expansions can not offer significant gains for cross-lingual consistency. Overall, the evaluation task does not inadequately boost consistency for languages using Latin script because word alignments resulting in cross-lingual consistency are the main factor.

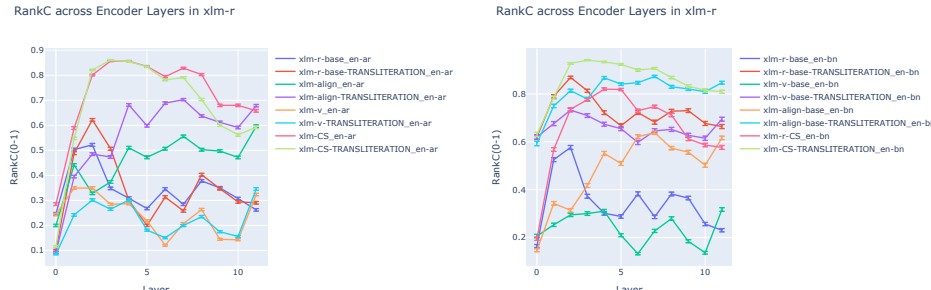

Figure 38: Impact of Transliterations.

---

[7]https://github.com/shhossain/BanglaTranslationKit
[8]https://github.com/hayderkharrufa/arabic-buckwalter-transliteration

