# OpenReview forum: "Is Knowledge in Multilingual Language Models Cross-Lingually Consistent?"
_ICLR.cc/2025/Conference — Submitted to ICLR 2025_

### Official Review · Reviewer_KcZx · 2024-10-31

**Soundness:** 2
**Presentation:** 2
**Contribution:** 3
**Rating:** 5
**Confidence:** 4

**Summary:**

This paper investigates the cross-linguistic consistency of multilingual models. To do this, the authors use the zero-shot cloze-task retrieval format enabled by the mLAMA dataset. However, the task is slightly modified because the prompts are mixed in several languages: the model must not complete the prompt “The capital of England is ___” but “The capital of Grande-Bretagne is ___”. So the words “England” and “Grande-Bretagne” here co-refer to the same underlying concept and should be treated identically by a cross-linguistically consistent model. To measure this consistency, the authors propose several metrics, such as accuracy at rank 1, weighted accuracy at rank 5, but also more precise metrics such as attention analysis or the IG² score.

The XLM-R and mT0 families of models are studied. The main findings are that the models possess a cross-linguistic consistency that is impacted by geographical proximity, language family and writing script. The authors then propose a more in-depth analysis, looking at the consistency score layer by layer and identifying a bottleneck in the intermediate layers.

Finally, the last part of this paper focuses on attempts to improve this consistency in the light of previous findings. To this end, the authors propose several techniques: biasing the model by removing attention or modifying FFN activations, but also by expanding the vocabulary or adding a supervised word alignment training objective. These last two methods are interesting as a direction for future work.

**Strengths:**

- The theme of this paper is very interesting and important.
- Many metrics are introduced, enabling in-depth analysis.
- Use of causal methods to improve and verify their findings a posteriori.

**Weaknesses:**

- Figure presentation: labels/titles should be larger, e.g. Figure 2 shows the same legend and title 3 times, which could be unified in the same graph (or as a table).
- The task definition section is understandable, but I think it needs some rework. In particular, it could benefit greatly from examples as notations are introduced. Finally, I had trouble understanding the baseline: here again, an illustrated explanation would be welcome.
- There are no error bars, yet the effects sometimes seem small (e.g. Figure 2). For example, the differences noted in Figure 5 are subjective and could be more clearly measured. Without clearer statistics, my conclusions from Figure 6 are opposed to those of the authors: I have the feeling that the interventions haven't changed anything, or even degraded consistency!
- Related work is a little light and could be fleshed out further

**Questions:**

1) Have you controlled for the fact that similar languages will sometimes have the same token for proper nouns, thus artificially skewing consistency? For example, “Paris” is spelled the same in French and English, but differently in Chinese.
2) Why not use the baseline for Figures 2, 7 and 10?
3) Why did you choose a code-mixed set-up? It's an out-of-distribution scenario, which can lead to strange behavior that's not representative of reality?
4) Why did you just look at the sum of the IG²s, thus losing information at neuron level, when you could have looked directly at whether the same neurons activated (using an overlap coefficient, for example)?

---

> ### Author Response · Authors · 2024-11-21
> **Thanks for feedback**
>
> **Weakness**
>
> 1. Figure 2 needs improvements.
>
> Thanks for the suggestion. We have recreated figures for the rebuttal revision. We would appreciate it if you can reassess this point.
>
> 2. Can you provide some input sample in helping with task definition.
>
> Thanks for this suggestion. In the rebuttal revision, the task definition is revised, and some input examples are added to Appendix (A.1).
>
> 3. Do you have error bars or complete statistics about Figure 2 or the overall performance?
>
> Yes, we have. We present the result in Appendix for each language. Please follow the link in the caption of each figure.
>
> 4. Do you have a statistical correlation for Figure 5?
>
> Yes, in Table 1, we provide statistical spearman correlation. We observe a moderate correlation with the cross-lingual consistency.
>
> 5. Can you explain the result of interventions for Figure 6?
>
> Following your instructions, we have added error bars to all results including Figure 6 for the rebuttal revision. We would appreciate it if you could reassess this point.
>
> We also want to clarify this. In our contribution list, we state “There is a partial causality (in a positive way) between adding monolingual biases and improving cross-lingual knowledge consistency. Thus, adding bias could be a potential method to calibrate consistency across languages.”. Specifically, in the discussion section (4.1), we confirmed that the interventions can modify the consistency but can only offer limited gains subject to mode size and architecture.
>
> 5. Related work is a little light and could be fleshed out further.
>
> Thanks for your suggestion. We have revised Related work. We would appreciate it if you could reassess this point.
>
> **Question**
>
> 1. Have you controlled for the fact that similar languages will sometimes have the same token for proper nouns, thus artificially skewing consistency?
>
> Yes, that’s the limitation of this work for now. We clarify this limitation on Page 15. The ideal setting would be to match any of the possible aliases of “Paris” and then match them together. But this is quite tricky to do further analysis.
>
> 2. Why not use the baseline for Figures 2, 7 and 10?
>
> Thanks for your valuable suggestions. We have added the average performance to these figures as the baseline.
>
> 3. Why did you choose a code-mixed setup?
>
> We want to clarify this. In this paper, we are looking at another facet of multilingual language models. As mentioned in the Introduction section (line 53), multilingual models are able to generalize across languages.  The setting in our paper enables the model to link the common knowledge of the language model across different preceding subjects because of cross-lingual generalization. To illustrate given “Paris is the capital of France” and another example “<paris-in-opther-language> is the capital of France”, if the model is indeed cross-lingually consistent, this means that the model incorporates the entity France to other equivalent subjects written in another language due to cross-lingual generalization. This is evident in our experiments on transliteration (c.f., A.4.4, new for rebuttal revision), code-switching training (c.f. section 4.4) , and cross-lingual supervision (c.f., section 4.3), where we show cross-lingually consistent is proportional to entity alignments across languages.
>
>
>
>
>
> 4. Why did you just look at the sum of the IG²s, thus losing information at neuron level, when you could have looked directly at whether the same neurons activated (using an overlap coefficient, for example)?
>
> IG^2 in fact measures the neuron-level information in the form of the attribution score of the particular neuron to the logit of a particular token. The consideration of this because it directly measures the effect of neurons on the logits. Thus, it is more causally justified. On the other hand, neuron overlapping that probably relies on an activation while more efficient might not be a good one as the activation score of neurons might introduce spurious correlation unless it is followed by subsequent causal intervention. So that might be a good alternative but due to time constraints, we will do this in future work.

---

### Official Review · Reviewer_MFk7 · 2024-11-01

**Soundness:** 3
**Presentation:** 3
**Contribution:** 2
**Rating:** 6
**Confidence:** 4

**Summary:**

The submission asks whether the masked language models encode factual knowledge consistently across different languages. In the experiments, the authors propose to use a popular world knowledge dataset: mLAMA. Specifically, they use prompts with subjects translated into German, Indonesian, Tamil, and Arabic. Then, they compare the output distributions of the models for prompts in different languages, interpreting similar ranking of tokens as evidence of knowledge consistency.

Subsequent analysis uses a range of interpretability methods to identify the layers and attention patterns that are important for knowledge consistency. Furthermore, the study shows that particular design choices: larger subword vocabulary and using word alignment objectives in multilingual training benefit consistency across languages.

**Strengths:**

- The work presents a clear-cut research question about multilingual knowledge consistency and proposes a relatively simple experimental setting to address it.

- The experimental results are followed by a detailed analysis to identify the components and attention mechanisms contributing to knowledge sharing across languages.


- The authors compare XLM-Roberta results with models of similar architecture but differing in specific aspects: XLM-V with a larger vocabulary and XLM-Align using word alignment objective. The survey concludes that these design choices improve cross-lingual consistency.

**Weaknesses:**

- The paper makes the wrong assumption about English and Indonesian being related languages. In fact, the languages are significantly typologically different, but they both are written with Latin script.
My suspicion is that the proposed experimental setting can inadequately boost consistency for languages using Latin script because named entities (from mLAMA) in such languages have the same or similar orthographical forms. To address the concern, the authors could analyze the effect of writing script on consistency in more detail and consider alternative task formulation, e.g., with the whole prompt translated (and transliterated) into the target language.
[The concern was addressed by the analysis of the transliteration's role added during the rebuttal period]

- Chaotic notation makes the setting hard to understand. Especially section 2 contains multiple instances of notation that are not properly defined (e.g., {...}^{l1} or I^{\S^{l1}}). The methodology description should be clarified, I suggest including a specific example from a dataset. (Improved in the rebuttal version).

- The work only considers masked language models, while most newer models are causal. I do not agree with the justification for discarding causal LMs referring to their “inherent hallucination problem” (sic). Please note that masked language models have also been shown to be prone to hallucinations, e.g. by Myaney, Narayan et al. 2020.

**Questions:**

- Why do you claim that scaling is not a promising strategy for consistency based on the observed bottlenecks in mid-upper layers?  I do not see evidence of that based on the experimental setting.


- How do you treat the cases when the prediction in the original language is wrong? Should we then assign a positive score to the model prediction that repeats wrong predictions in another language?

---

> ### Author Response · Authors · 2024-11-21
> **Thanks for feedback**
>
> **Weakness**
> 1. Is Indonesian similar to English?
>
> As pointed out in the paper, we assume two languages are similar not necessarily have to be similar typologically because the grouping factor takes into account the cultural aspect that is less relevant for this task setting. Both languages share many commonalities in terms of grammar and writing scripts. We clarify this with reference in subsection 2.3.
>
> 2. Do you have more studies about the writing system?
>
> For the latin script, we acknowledge that this setting could make the evaluation biased towards languages written in latin script but this is still not sufficient to cover many of the languages and we still need to figure this out. Another issue of transliteration is the orthography inconsistency.
>
> In addition, we include a limitation section in our paper. We clarify this in our task definition.
>
> 3. Why do you consider masked language models?
>
> Thanks for the suggestion and reference. We would like to clarify this. We are trying to say “less hallucination”, which is supported by [1, 2]. Due to the time constraint, we can not investigate causal LM during rebuttal. We want to add additional experiments to the next revision.
>
> [1] Fu, Zihao, et al. "Decoder-only or encoder-decoder? Interpreting language model as a regularized encoder-decoder." arXiv preprint arXiv:2304.04052 (2023).
>
> [2] Ziwei Ji, Nayeon Lee, Rita Frieske, Tiezheng Yu, Dan Su, Yan Xu, Etsuko Ishii, Ye Jin Bang, Andrea Madotto, and Pascale Fung. 2023. Survey of Hallucination in Natural Language Generation. ACM Comput. Surv. 55, 12, Article 248 (December 2023), 38 pages. https://doi.org/10.1145/3571730
>
> 4. Notations need improvements.
>
> Thanks for your suggestions. We have rewritten the task definition with improved notations for the rebuttal revision. We will appreciate it if you can reassess this point.
>
> **Question**
>
> 1. Can you justify the claim that scaling is not a promising strategy?
>
> We scaled up the experiment to larger models (from 0.3B  to 10B) and observed quite similar patterns across layers (i.e., the consistency does not increase monotonically but vacillates till the last layer). The typical pattern is presented in Figure 3. In the rebuttal revision, we clarify this around line 200 as “To better understand the cross-lingual consistency bottleneck, we examine the layer-wise consistency patterns across different model sizes, as presented in Figure 3”.
>
> 2. How do you treat the cases when the prediction is wrong?
>
> We would like to justify our task definition. This study focuses on consistency of knowledge across multiple languages. We want to know the ability of multilingual language models to recall the same knowledge no matter which languages are written in the context. So its different evaluation of cross-lingual accuracy and this aspect is not a standalone aspect of cross-lingual transfer. We clarify this in the task definition for the rebuttal revision as “Note that we do not consider whether the prediction is correct. Instead, $f_{metric}$ evaluates the parity and consistency across the languages that the model is expected to output similar candidates for $I_{mono}$ and $I_{cm}$.

---

> > ### Comment · Reviewer_MFk7 · 2024-11-21
> >
> > Thank you for your response and for submitting an updated version. Section 2 is now clearer thanks to the inclusion of additional definitions and helpful examples.
> >
> > I have updated the review and increased the score for presentation.

---

> > > ### Author Response · Authors · 2024-11-22
> > >
> > > Thanks for your extra effort to re-assess our revision. Please let us know  if you still have any additional concerns, questions, or actionable suggestions.

---

> ### Author Response · Authors · 2024-11-25
>
> Weakness:
>
> - My suspicion is that the proposed experimental setting can inadequately boost consistency for languages using Latin script because named entities (from mLAMA) in such languages have the same or similar orthographical forms. To address the concern, the authors could analyze the effect of writing script on consistency in more detail and consider alternative task formulation, e.g., with the whole prompt translated (and transliterated) into the target language.
>
> Thanks for this insightful comment. We follow your intructions and add a new section to Appendix (A 4.4) to study transliteration.
>
> Instead of using translations, we transliterate bn and ar to understand the impact of writing systems, particularly transliterations. As presented in Figure 38, word alignments (or the similar effect from CS training) contribute to the model’s cross-lingual consistency against writing systems because xlm-align and xlm-r-cs show similar performance in both original and transliteration settings. Meanwhile, we can observe that xlm-align and xlm-r-cs significantly improve the overall performance for non-Lattin scripts in section 4.3. This is reasonable as word alignments or CS training help the model link original words with their translations or transliterations, depending on the training corpus, thereby enhancing cross-lingual consistency. We suspect that these word alignments might also improve robustness for handling non-standard spellings and orthographic variations. However, xlm-v-base and xlm-r-base without word alignment benefit from transliterations, which means that xlm-v-base and xlm-r-base do not sufficiently align original words with their transliterations to main cross-lingual consistency. It is also confirmed by the overall performance of vocabulary expansions in section 4.2 , where vocabulary expansions can not offer significant gains for cross-lingual consistency. Overall,  the evaluation task does not inadequately boost consistency for languages using Latin script because word alignments resulting in cross-lingual consistency are the main factor.
>
> (May help you recall the paper: 1) xlm-align: xlm-r-base + word-level cross-lingual supervision; 2) xlm-v-base: xlm-r-base with expanded vocabulary; 3) xlm-r-cs:  xlm-r-base with code-switching training)

---

> > ### Comment · Reviewer_MFk7 · 2024-11-26
> >
> > The additional analysis, to a significant extent, addresses my concern about the role of script sharing. Thank you for adding it, and please comment on these findings in the main text as well.
> >
> > I'm happy to increase my overall score after considering the responses from the authors.

---

> > > ### Author Response · Authors · 2024-11-27
> > >
> > > Thank you for your reassessment and time.

---

### Official Review · Reviewer_brtt · 2024-11-04

**Soundness:** 3
**Presentation:** 4
**Contribution:** 3
**Rating:** 8
**Confidence:** 3

**Summary:**

The paper explores knowledge consistency across languages in multi-modal encoder type LLMs by testing on a cloze task with  parallel triples in 52 languages focused on referencing tasks. The paper uses several metrics: consistency metrics, consistency evolution, subset-object attention, and a slightly modified version of a metric called IG^2 to measure levels of cross-lingual consistency. The paper finds that multi-lingual models show different levels of consistency depending on the "differences" among pairs of languages, based on geographical distance, language family and use of similar or different scripts. To obtain better consistency in knowledge across languages, they experiment with some mitigation techniques and find that adding monolingual examples to bias, and adding multi-lingual vocabulary and pre-training with aligned words improve results.

**Strengths:**

Knowledge consistency across multiple languages is important if multi-lingual language models are to be used with confidence in tasks across languages. This paper uses existing metrics to compute knowledge inconsistency in language pairs. The results clearly show that language family and the scripts used matter in the measures of consistency. Similar languages show more consistency, and dissimilar languages shows less consistency. Languages using similar scripts, (i.e., likely similar family and geographical close languages) show higher consistency also. The paper also develops ways to reduce inconsistent knowledge using monolingual biases and appropriate pre-training.

**Weaknesses:**

The fact that knowledge consistency across languages is higher if they are in the same language group or use the same script seems somewhat self-evident, although it is nice to find experimental evidence for the same. The paper works with one multi-lingual reference-oriented code-switched dataset. It also experiments with only one encoder-only model, although two sizes of it.

**Questions:**

What are the types of references the dataset has? Some examples will be nice. Do some reference types work better than others? Some examples will be nice in the appendix. What is the alignment objective? Is there no need for analysis of subject-verb attention  and/or object-verb attention? Can any meaningful comparison be performed with methods published in other relevant papers?

---

> ### Author Response · Authors · 2024-11-21
> **Thanks for feedback**
>
> **Weakness**
>
> 1. The paper only experiments with encoder models.
>
> We would like to clarify this. We also evaluated encoder-decoder models, mt0 (base, large, and xl). Please refer to subsection 2.4. Meanwhile, following your suggestions, we add our preliminaries in comparison between mt0 and mt5 to subsection 4.5 to investigate the effectiveness of multilingual multi-task fine-tuning.
>
> **Question**
>
> 1. Do you have any study about the correlation between types of references and cross-lingual consistency?
>
> Thanks for this interesting question. We have not investigated consistency from this angle. We want to add this analysis to our future work list.
>
> 2. What is the alignment objective in xlm-r-align?
>
> xlm-r-align [1] trains the model with word-to-word alignment objectives.
>
> 3. Why do you consider subject-verb attention and/or object-verb attention?
>
> For attention, we think that intuitively both analyses are not that crucial for this setting because we only measure how the model binds the relation of two entities on which one entity that we assume as common knowledge to another entity that is related to that common knowledge across languages.
>
>
> [1] Zewen Chi, Li Dong, Bo Zheng, Shaohan Huang, Xian-Ling Mao, Heyan Huang, and Furu Wei. 2021. Improving Pretrained Cross-Lingual Language Models via Self-Labeled Word Alignment. In Proceedings of the 59th Annual Meeting of the Association for Computational Linguistics and the 11th International Joint Conference on Natural Language Processing (Volume 1: Long Papers), pages 3418–3430, Online. Association for Computational Linguistics.

---

### Official Review · Reviewer_aHwz · 2024-11-04

**Soundness:** 3
**Presentation:** 3
**Contribution:** 2
**Rating:** 5
**Confidence:** 4

**Summary:**

This paper investigates cross-lingual factual consistencies for language models, in particular for multilingual encoder-only models like XML-r and multilingual encoder-decoder models like mT0.

Their investigation proceeds along a number of dimensions, including distances between languages as well as a number of mitigation strategies. While the mitigation strategies offer some improvements, cross-lingual supervision directly leads to the largest benefits.

**Strengths:**

Addressing cross-lingual factual consistency is an important problem, which is investigated from several angels.

**Weaknesses:**

My main problem with this paper is the choice to trigger/prompt cross-lingual knowledge through code-switching. This seems a very unnatural choice, given that the models were never exposed (unless by coincidence) to code-switched data. I have serious doubts that the outcomes tell us a lot about the cross-lingual capabilities of these models. Why not use Translations of cloze prompts to measure consistency? I am aware of the fact that code-switching has been used to train multi-lingual models (e.g., mRASP) and that it is beneficial to induced shared representations across languages, but I don't see why this would sense for evaluation.

**Questions:**

While you look into differences between languages, to what extent does the amount of data used during pre-training impact cross-lingual consistency?

---

> ### Author Response · Authors · 2024-11-21
> **Thanks for feedback**
>
> **Weakness**
>
> 1. Why not use Translations of cloze prompts to measure consistency?
>
> Thanks for the valuable comments. Yes, using translations is an intuitive way to evaluate multilinguality and cross-linguality. In this paper, we are looking at another facet of multilingual language models. As mentioned in the Introduction section (line 53), multilingual models are able to generalize across languages.  The setting in our paper enables the model to link the common knowledge of language model across different preceding subjects because of cross-lingual generalization. To illustrate given “Paris is the capital of France” and other example “<paris-in-opther-language> is the capital of France”, if the model is indeed cross-lingually consistent, this means that the model incorporates the entity France to other equivalent subjects written in another language.
>
> 2. How about including code-switching corpus?
>
> Thanks for suggesting a model with an explicit code-switching objective in training. However, the suggested model is a multilingual translation model trained on parallel datasets only which is different from the multilingual language models trained on the large-scale monolingual corpus. Alternatively, we find an open-source model [1] fine-tuned from XLM-r on entity-centric code-switching data. The result also aligns with our findings. We have added the result to our rebuttal revision. Please refer to subsection 4.4. Due to time constraints, we can not provide the overall consistency results for this model. We will add it to our next revision.
>
> **Question**
>
> 1. While you look into differences between languages, to what extent does the amount of data used during pre-training impact cross-lingual consistency?
>
> Thanks for contributing an interesting question.  While this factor is not in the scope of this paper, we can still get some clues from our findings. Speccifcally, using a multilingual multi-task corpus cannot offer significant gains to cross-lingual consistency (section 4.5). However, using code-swithcing dataset or alignments are promising. In conclusion, the benefit of using additional dataset highly depends on the information of that dataset (section 4.4).
>
> We also want to share an existing work [2] to support us. This paper claims “The moderate correlation indicates a limited impact of the training data volume on learning factual knowledge”.
>
> [1] Chenxi Whitehouse, Fenia Christopoulou, and Ignacio Iacobacci. 2022. EntityCS: Improving Zero-Shot Cross-lingual Transfer with Entity-Centric Code Switching. In Findings of the Association for Computational Linguistics: EMNLP 2022, pages 6698–6714, Abu Dhabi, United Arab Emirates. Association for Computational Linguistics.
>
>
> [2] Xin Zhao, Naoki Yoshinaga, and Daisuke Oba. 2024. Tracing the Roots of Facts in Multilingual Language Models: Independent, Shared, and Transferred Knowledge. In Proceedings of the 18th Conference of the European Chapter of the Association for Computational Linguistics (Volume 1: Long Papers), pages 2088–2102, St. Julian’s, Malta. Association for Computational Linguistics.

---

> > ### Comment · Reviewer_aHwz · 2024-11-27
> >
> > Thank you for the clarifications. While I understand the emphasis on cross-linguality, the same can also be achieved with translated prompts, where the prompt is in language A and has to be matched with facts expressed in the training data in language B. As I've mentioned, this would be a much more realistic scenario. Nevertheless, I appreciate the other answers and raised my soundness score.

---

> ### Author Response · Authors · 2024-11-25
>
> Thanks for the code-switching suggestion.
>
>  We add a new section to Appendix (A 4.4) to study transliteration. We additionally find code-switching training contributions to robustness for handling transliterations and non-standard spellings.

---

> ### Author Response · Authors · 2024-11-27
>
> Thanks for your feedback and time. We want to clarify and discuss our design.
>
> Indeed, parallel pairs can be used for our evaluation task. However, we have one major concern about monolingual training bias in cross-lingual consistency. Multilingual language models are trained with the monolingual objective, i.e., the input and output languages are the same. However, the language modeling performance varies across languages due to the training corpus. The output distribution of knowledge recall is biased by the monolingual training objective (c.f., section 2). This might impact our analysis regarding distribution differences. Conversely, we mitigate bias in our code-mixed context-independent setting by maintaining the language surface—only altering the subject without changing the overall structure—and the language domain.  This is evident in our transliteration experiments (c.f., A.4.4, new for rebuttal revision) as quoted below:
> ``` Instead of using translations, we transliterate bn and ar to understand the impact of writing systems, particularly transliterations. As presented in Figure 38, word alignments (or the similar effect from CS training) contribute to the model’s cross-lingual consistency against writing systems because xlm-align and xlm-r-cs show similar performance in both original and transliteration settings. ........... However, xlm-v-base and xlm-r-base without word alignment benefit from transliterations, which means that xlm-v-base and xlm-r-base do not sufficiently align original words with their transliterations to main cross-lingual consistency. ...... Overall, the evaluation task does not inadequately boost consistency for languages using Latin script because word alignments resulting in cross-lingual consistency are the main factor. ```
>
> In conclusion, if coreferential subject entries are trained to generalize across languages (e.g., xlm-align and xlm-r-cs), we could observe the cross-lingual consistency that is not significantly biased by the monolingual training objective, due to the preserved language surface in code-mixing and cross-lingual generalization. If not, the cross-lingual consistency will be weak. Additionally, utilizing two parallel sentences in different languages tends to be influenced by the monolingual training objective, even after coreferential subject entries have been aligned.
>
> (May help you recall the paper: 1) xlm-align: xlm-r-base + word-level cross-lingual supervision; 2) xlm-v-base: xlm-r-base with expanded vocabulary; 3) xlm-r-cs: xlm-r-base with code-switching training)
>
> We also want to clarify our goal.  We try to evaluate the hypothesis: multilingual language models recall consistent factual knowledge for coreferential statements in cross-lingual settings, and we answer two key questions: 1) do multilingual language models recall factual knowledge for the coreferential statements in a similar manner on different languages? and 2) how does the mechanism of multilingual language models work on the incorporation between entities or references in cross-lingual settings?    To address these questions, we create code-mixed coreferential statements from monolingual statements by substituting a subject entity with an equivalent one in another language that shares the same reference, rather than directly analyzing parallel sentences where each statement recalls entities or references within the same language. This approach allows us to understand multilingual knowledge recall in a cross-lingual setting.

---

### Author Response · Authors · 2024-11-21
**General**

General:

Thanks for all the valuable comments. We have uploaded a rebuttal revision.

Major Updates:
1. We polish and proofread the paper based on comments.
2. Following reviewers KcZx and MFk7, we clarify task definitions and provide our input templates (sec 2.1).
3. We address typos in Table 1.
4. Following reviewer aHwz, we analyze a new model fine-tuned on code-mixed dataset (sec 4.4). Overall, code-switching training offer significant gains.
5. Following reviewer brtt, we provide our preliminaries on mt5 in comparison to mt0, which can be used to discuss the potential mitigation for encoder-decoder models. Overall,  mt5 and mt0 are similar in cross-lingual consistency.
6. Following reviewer MFk7, we add a new section to Appendix (A 4.4) to study transliteration. Overall,  the evaluation task does not inadequately boost consistency for languages using Latin script because word alignments resulting in cross-lingual consistency are the main factor in the experiments.
6. Following KcZx,  we add error bars to all figures.

Due to time constraints, we only add results on 4 similar and dissimilar languages for new models suggested by reviewers. We will add the overall result (based on the three factors) to our next revision, which requires us to run experiments for all 53 languages.

We hope our rebuttal revision can mitigate the concerns. Further questions and comments are always welcome.

---

### Comment · Area_Chair_LdEG · 2024-11-25
**Respond  to the author's rebuttals**

Dear Reviewers,

Thank you for your efforts in reviewing this paper. We strongly encourage you to review and respond to the author's comments to promote a more dynamic exchange of ideas.

Thank you for your collaboration.

---

### Author Response · Authors · 2024-11-30
**Near Deadline**

Dear reviewers,

Thank you very much for your time. If you have future questions, please let us know. We would appreciate it if you could reassess our paper based on our responses and the rebuttal revision.

---

### Meta-Review · Area_Chair_LdEG · 2024-12-19

**Metareview:**

The paper investigates cross-lingual factual consistency in multilingual language models, focusing on encoder-only models like XML-R and encoder-decoder models like mT0. The study examines how these models maintain knowledge consistency across different languages, using a cloze task with parallel triples in 52 languages, leveraging the mLAMA dataset.

However, there are noteworthy concerns. 1)  The novelty of the paper is limited, as the cross-Lingual consistency of factual knowledge in pre-trained language models has been studied in previous works.  2) Clarifying the motivation behind using code-switching to trigger or prompt cross-lingual knowledge is essential. 3) Including experiments on decoder-based language models would further strengthen and validate the conclusions.

Addressing the aforementioned questions will render the paper more comprehensive and impactful.

**Additional Comments On Reviewer Discussion:**

As demonstrated in the 'General' in official comment, the authors have refined the paper to address and alleviate the reviewers' concerns.

---

### Decision · Program_Chairs · 2025-01-22

Reject